# Global hypo-methylation in a proportion of glioblastoma enriched for an astrocytic signature is associated with increased invasion and altered immune landscape

James Boot[1], Gabriel Rosser[1], Dailya Kancheva[2], Claire Vinel[1], Yau Mun Lim[3], Nicola Pomella[1], Xinyu Zhang[1], Loredana Guglielmi[1], Denise Sheer[1], Michael Barnes[4], Sebastian Brandner[3], Sven Nelander[5], Kiavash Movahedi[2], Silvia Marino[1]*

[1]Blizard Institute, Barts and The London School of Medicine and Dentistry, Queen Mary University, London, United Kingdom; [2]Laboratory for Molecular and Cellular Therapy, Vrije Universiteit Brussel, Brussels, Belgium; [3]Division of Neuropathology, The National Hospital for Neurology and Neurosurgery, University College London Hospitals NHS Foundation Trust, and Department of Neurodegenerative Disease, Queen Square, Institute of Neurology, University College London, London, United Kingdom; [4]Centre for Translational Bioinformatics, William Harvey Research Institute, Barts and The London School of Medicine and Dentistry, Queen Mary University of London, London, United Kingdom; [5]Department of Immunology Genetics and Pathology, Uppsala University, Uppsala, Sweden

*For correspondence:
s.marino@qmul.ac.uk

Competing interest: The authors declare that no competing interests exist.

**Abstract** We describe a subset of glioblastoma, the most prevalent malignant adult brain tumour, harbouring a bias towards hypomethylation at defined differentially methylated regions. This epigenetic signature correlates with an enrichment for an astrocytic gene signature, which together with the identification of enriched predicted binding sites of transcription factors known to cause demethylation and to be involved in astrocytic/glial lineage specification, point to a shared ontogeny between these glioblastomas and astroglial progenitors. At functional level, increased invasiveness, at least in part mediated by SRPX2, and macrophage infiltration characterise this subset of glioblastoma.

## Editor's evaluation

Overall, this paper represents a large amount of highly detailed work that identifies a hypomethylated subtype of glioblastoma that provides exciting foundations for further translational investigations.

## Introduction

Glioblastoma IDH-wildtype (now renamed glioblastoma) is a highly aggressive brain tumour (*Wen and Kesari, 2008*), with an extremely poor prognosis of 15 months' median survival from diagnosis (*Alcantara Llaguno and Parada, 2016*). Glioblastoma is also the most prevalent primary adult brain tumour with an annual occurrence of approximately five cases per 100,000 people (*Wen and Kesari, 2008*; *Omuro and DeAngelis, 2013*) and a mean diagnosis age of 64 (*Bush et al., 2017*).

Part of the difficulty in researching and treating glioblastoma is the heterogeneity of the tumour, both at inter- and intra-tumoural levels. Inter-tumoural heterogeneity reflects differences at the (epi) genetic level which is illustrated by the classification of glioblastomas into different subgroups, such as the transcription-based subgrouping proposed by the Verhaak group (*Wang et al., 2017a*), and the DNA methylation-based subgrouping proposed by *Sturm, 2012*. However, seminal studies by multiple groups have established that there is also considerable intra-tumoural heterogeneity in glioblastoma (*Patel et al., 2014*; *Neftel et al., 2019*; *Couturier et al., 2020*). Established transcriptional glioblastoma subgroups profiles (Proneural, Neural, Classical, and Mesenchymal) are variably expressed in single cells from the same tumours (*Patel et al., 2014*) and tumour cells are characterised by distinct gene expression signatures and cluster separately from one another (*Neftel et al., 2019*). Single cells have also been shown to score highly for multiple gene expression signatures creating hybrid states, which further increases the heterogeneity of the tumour cell populations (*Neftel et al., 2019*). *Couturier et al., 2020*, found that tumour cells fell into a spectrum of states, whereby cells at one end of the spectrum highly expressed neuronal genes, whilst cells at the other end of the spectrum up-regulated astrocytic genes. Subgrouping based on DNA methylation subtyping alone (*Capper et al., 2018*) show less heterogeneity (*Sturm, 2012*), although some degree of heterogeneity has been reported in a small number of cases (*Wenger et al., 2019*). The tumour microenvironment also contributes to glioblastoma heterogeneity and the crosstalk between malignant cells and for example the inflammosome is well characterised (*Daubon et al., 2020*) with macrophages known to drive the transition of cancer cells towards a mesenchymal-like state (*Hara et al., 2021*).

Whether glioblastoma heterogeneity and its underlying epigenetic makeup is determined by the cell of origin or is acquired during transformation is a matter of debate. The putative cell of origin is thought to be a stem/progenitor cell that acquires the first genetic and/or epigenetic alterations that promote the formation of the tumour (*Alcantara Llaguno and Parada, 2016*). The debate over the cell of origin in glioblastoma centres around neural stem cells (NSCs) and lineage committed progenitor cells, such as oligodendrocyte, astrocytic, and neuronal precursor cells. NSCs are logically the prime candidate for the cell of origin of glioblastoma, because of their self-renewal potential, differentiation plasticity, and similarity in their gene expression with glioblastoma stem cells (here called glioblastoma initiating cells [GICs]) (*Fan et al., 2019*). A seminal study by *Zheng et al., 2008*, in mice, showed that deletion of both *Trp53* and *Pten* by Cre recombinase under the control of the *GFAP* promoter increased the proliferative rate and self-renewal capability of NSCs, whilst also inhibiting their ability to differentiate into specific neural lineages, leading to transformation into high-grade malignant gliomas. Importantly, glioblastoma driver mutations have been identified in NSCs of the human subventricular zone (SVZ) in tissue samples obtained from patients, providing for the first-time evidence that these cells can act as cells of origin of glioblastoma in human (*Lee et al., 2018*). Further mouse studies (*Alcantara Llaguno et al., 2015*) have shown that inducing null alleles of *Nf1*, *Trp53*, and *Pten* using Cre recombinase under the control of NSC-specific *Nestin*, but also oligodendrocyte progenitor cell (OPC)-specific *NG2* or bipotential progenitor cell-specific *Ascl1* led to high-grade glioma, whilst no tumours formed when the same null alleles were induced in mature neurons (*Camk2a-Cre*), immature neurons (*Neurod1-Cre*), and adult neuronal progenitors (*Dtx1-Cre*) (*Alcantara Llaguno et al., 2019*). These studies together with other mouse studies have also shown that OPC (*Liu et al., 2011*; *Lu et al., 2016*) and astrocyte precursors (*Chow et al., 2011*) can behave as cell of origin of glioblastoma raising the possibility that tumours may originate from glial progenitor cells at different developmental stages, hence contributing to their heterogeneity.

Here, we leveraged pairs of patient-derived GICs and patient-matched expanded potential stem cells (EPSCs)-derived neural stem (*Vinel et al., 2021*) and progenitor cells to investigate the DNA methylation landscape of GICs as compared to their putative cell of origin in a patient-specific manner to further characterise glioblastoma heterogeneity and its ontogeny.

## Results
### A DNA hypo-methylation bias in a subset of GBM

We have recently described the generation of a cohort of 10 pairs of GICs and syngeneic induced NSCs (iNSCs), where the latter has been validated as a suitable proxy for endogenous NSCs at both transcriptional and epigenetic level (*Vinel et al., 2021*). Here, we have leveraged this resource (*Vinel*

*et al., 2021*) to investigate the DNA methylome of GICs as compared to syngeneic iNSCs. DNA methylation was assessed using the Illumina Infinium Methylation EPIC array and data processed as described in the Materials and methods. Across the 10 intra-patient comparisons, we visualised the distribution of the median probe delta M values and the proportion of these probes that were either hypo- or hyper-methylated, as well as the number and proportion of hypo- and hyper- methylated regions (differentially methylated regions [DMRs]) (*Figure 1A*, left panel). GICs 19, 30, 31, and 17 stood out as they had over 60% hypo-methylated DMRs, and together had a statistically significant larger proportion of hypo-methylated probes and DMRs as compared to the other six GICs in the cohort. These four GICs were identified as 'hypo-bias' – herein referred to as bias-GICs (B-GICs) (*Figure 1A*, left panel and *Figure 1B*). This subgrouping did not reflect the known DNA methylation-based classification of *Sturm, 2012*; three of the four B-GICs belonged to the RTKI (Proneural) subgroup and one to the RTKII (Classical) subgroup. The GICs without this hypo-bias, herein referred to as non-B-GICs (nB-GICs) were spread across the RTKI, RTKII, and Mesenchymal subgroups. Noticeably, the proportion of hypo-methylated and hyper-methylated DMRs was exaggerated, when patient-specific probes and DMRs were considered (*Figure 1A*, right panel).

Principal component analysis (PCA) of the 5000 most variable DNA methylation probes across all GICs in our cohort showed that principal component 1 (PC1) largely separated GICs by hypo-methylation bias, with three out of four B-GICs (19, 30, and 31, highlighted) clustering together on the left of the plot and all remaining GICs to the right (*Figure 1C*). A heatmap dendrogram of the beta values of the top 100 probes driving PC1 showed that these probes had much lower beta values in B-GICs 19, 30, and 31 relative to all other GICs, reflecting the separate cluster observed in the PCA (*Figure 1D*). This observation suggested that the hypo-methylation bias was GIC-driven, and not caused by the iNSC comparator, a conclusion which was confirmed when we performed non-syngeneic comparisons between each GIC and each of the iNSCs in our cohort (*Figure 1E*). GIC17 was found to be an exception as the proportion of hypo-methylated DMRs was not as high as the other three GICs (19, 30, and 31), and indeed this GIC did not cluster with GICs 19, 30, and 31 in the PCA plot (*Figure 1C*). However, the average percentage of hypo-methylated DMRs for this GIC was still greater than 60% regardless of the comparator used. Interestingly, the average percentage of hypo-methylated DMRs for GIC26 was also greater than 60%, despite the cell line not meeting this threshold in the syngeneic comparison, possibly due to significant variability of the two biological replicates (*Figure 1—figure supplement 1:*). The proportion of hypo-methylated DMRs for the remaining GICs varied from 20% to 70%, however the mean percentage of hypo-methylated DMRs was below threshold (60%).

To validate this observation in an independent GIC cohort, DNA methylation data from the publicly available HGCC resource (*Xie et al., 2015*) was used. The HGCC dataset contains 71 GIC samples and given that non-syngeneic comparisons do not prevent the identification of the hypo-methylation bias (*Figure 1E*), these GICs were compared with iNSCs from our cohort. HGCC GIC lines underwent differential methylation comparisons against four different iNSC lines in turn to determine whether the results were consistent across comparisons. The reported percentage of hypo-methylated DMRs was found to be consistent regardless of the iNSC comparator used (*Figure 1G*), in keeping with the results of the non-syngeneic comparisons performed in our own cohort (*Figure 1E*). Given that the percentage of hypo-methylated DMRs in each GIC was determined four times – once for each iNSC comparator, each GIC had to meet a more lenient threshold of 50% hypo-methylated DMRs in all five iNSC comparisons in order to be classified as a B-GIC. Using this threshold, 46.5% of HGCC GICs were found to have a hypo-methylation bias. This was comparable with the proportion in our smaller cohort where we found 5/11 (45.5%) had greater than <50% DMRs and 4/11 (36.4%) had <60% DMRs. Next, for each HGCC GIC, the mean percentage of hypo-methylated DMRs when compared against all iNSCs was determined, to assess the extent of hypo-methylation bias for that GIC. HGCC GICs were stratified into a spectrum of states of either having no hypo-methylation bias (<50% hypo-methylated DMRs), a very low bias (>50%), low bias (>60%), medium bias (>70%), high bias (>80%), or very high bias (>90%). PCA of the 5000 most variable methylation probes across the HGCC GICs showed that PC1 separated patients on the extent of their hypo-methylation bias with very high B-GICs clustering to the right, and those with no bias clustering to the left (*Figure 1F*). TCGA subgroup classification confirmed no enrichment for a specific subgroup (of the 33 GICs deemed to have a hypo-methylation bias, 15 were Mesenchymal, 4 Proneural, 6 Classical, and 8

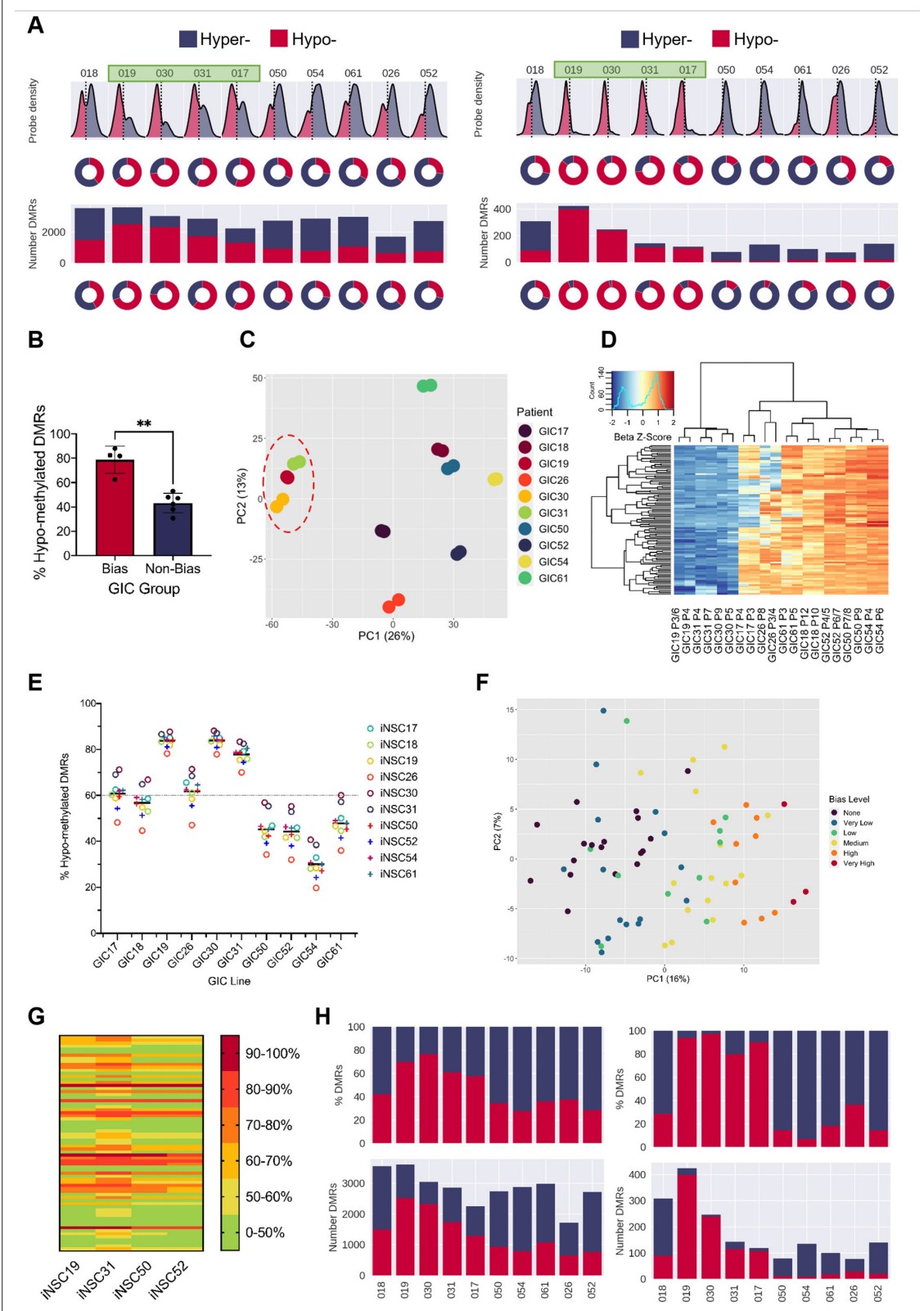

**Figure 1.** A DNA hypo-methylation bias in a subset of GBM. (**A**) The distribution of median probe delta M values (first row), the proportion of hypo- (red) and hyper- (blue) probes (second row), total number of differentially methylated regions (DMRs) (third row), and the proportion of hypo- and hyper- DMRs (fourth row) for each glioblastoma initiating cell (GIC)-induced neural stem cell (iNSC) comparison (left panel) and using patient-specific probes and DMRs (right panel). Samples with a hypo-methylation bias highlighted in green. (**B**) Percentage of hypo-methylated DMRs in bias-GICs (N=4) and

*Figure 1 continued on next page*

*Figure 1 continued*

non-bias-GICs (N=6), statistical significance tested using Welch's t-test. (**C**) Principal component analysis (PCA) of all patient-derived GIC samples from our cohort, based on the top 5000 most variable methylation probes, GICs 19, 30, and 31 highlighted by red circle. (**D**) Heatmap dendrogram of the beta value z-scores of the top 100 methylation probes driving the variation of PC1 across all GICs and replicates in our cohort. (**E**) Percentage of hypo-methylated DMRs from all possible comparisons of GICs and iNSCs in our cohort, the mean percentage hypo-methylated DMRs for each GIC is represented by a horizontal black line. (**F**) PCA of all patient-derived GIC samples from the HGCC cohort, based on the top 5000 most variable methylation probes. (**G**) Heatmap summary of the percentage of hypo-methylated DMRs from all possible comparisons of HGCC GICs and five iNSCs from our cohort. (**H**) Number (bottom panel) and proportion (top panel) of hypo- and hyper-DMRs for each patient comparison between formalin fixed paraffin embedded (FFPE) bulk tumour and iNSC, repeated in the right panel for patient-specific DMRs from the same comparisons.

The online version of this article includes the following source data and figure supplement(s) for figure 1:

**Source data 1.** Source data for *Figure 1*.

**Figure supplement 1.** Hypo-methylation bias does not globally impact on transcription, but bias-glioblastoma initiating cells (B-GICs) are transcriptionally different from non-B-GICs (nB-GICs).

**Figure supplement 1—source data 1.** Source data for *Figure 1—figure supplement 1*.

Neural), in keeping with the interpretation that the hypo-methylation bias observed is spread across the known glioblastoma subgroups.

Finally, formalin fixed paraffin embedded (FFPE) tumour tissue, which was available for all 10 GIC/iNSC pairs of our cohort, was used as the neoplastic comparator to exclude that the observed hypo-methylation bias was induced in GIC by in vitro culture. Once again, tumours 19, 30, 31, and 17 showed a greater proportion of hypo-methylated DMRs than the remaining six comparisons (*Figure 1H*, left panel), and the proportion of hypo-methylated DMRs increased when only considering patient-specific DMRs (*Figure 1H*, right panel).

In conclusion, we have identified a subset of glioblastoma that harbour a DNA hypo-methylation bias when either the bulk tumour or GIC derived thereof are compared to iPSC-derived NSC.

## B-GICs are transcriptionally distinct from nB-GICs

We used matched transcriptomic data from our GICs to assess any impact on transcription of the hypo-methylation bias. PCA based on the top 20% most variably expressed genes recapitulated the grouping of GICs observed at DNA methylome level (*Figure 1—figure supplement 1B*) with PC1 largely distinguishing the B-GIC and nB-GIC groups. Three out of four of the B-GICs (17, 19, and 30) grouped together to the right of the PC1 along with one biological replicate from GIC26 – a line characterised by substantial variability between replicas (*Figure 1—figure supplement 1A*). This grouping of GICs 17, 19, 26, and 30 did not correspond to their transcriptional GBM subgroup as they are RTKII, RTKI, MES, and RTKI, respectively. The remaining nB-GICs largely clustered together to the left of the PC2, with one of the B-GICs – GIC31 – lying between these two groups. Despite these differences at transcriptome level, patient-matched pairs of GICs and iNSCs did not reveal a bias in the directionality of differentially expressed genes (DEGs) (*Figure 1—figure supplement 1C*), indicating that the hypo-methylation bias does not translate to a bias in gene up- or down-regulation. Known genes associated with epigenetic remodelling or DNA methylation such as DNMTs or TETs were not identified within the DEGs (*Supplementary file 1*).

Next, we queried whether the hypo-methylation bias impacted miRNA expression and carried out small RNAseq. Clustering of the 500 most variably expressed miRNAs did not recapitulate the grouping of samples into B-GICs and nB-GICs (*Figure 1—figure supplement 1D*). Similarly, comparisons between GICs and iNSCs did not reveal a bias in directionality of expression of differentially expressed miRNAs (*Figure 1—figure supplement 1E*). Interestingly though, we identified five miRNAs differentially expressed in the same direction in all B-GICs (*Figure 1—figure supplement 1F*). Some of these five differentially expressed miRNAs have previously been linked either directly or indirectly to neural development and/or glial lineage specification. Silencing of miR-1275 induces *GFAP* (an astrocyte marker) expression in glioblastoma cells (*Mai et al., 2019*), and its down-regulation was associated with oligodendroglia differentiation of tumour cells (*Katsushima et al., 2012*) – similarly in our dataset this miRNA is down-regulated. The *JAK/STAT* pathway, which is known to be a key regulator of astrocyte differentiation and activation (*Xiao et al., 2010*; *He et al., 2005*; *Bonni et al., 1997*; *Ceyzériat et al., 2016*), is thought to be up-regulated by miR-4443, which is down-regulated in our GICs. Finally, miR-196, which is up-regulated in B-GICs, plays an essential role in neural development

as it helps regulate Homeobox (*HOX*) genes (*Sehm et al., 2009*), some of which have been shown to play a role in astrocyte biology (*Vivinetto et al., 2020*).

In summary, the hypo-methylation bias identified here did not lead to a bias in up- or down-regulation of gene or miRNA expression at a global level. However, PCA of the most variable genes indicated that there were transcriptional and regulatory differences between B-GICs and nB-GICs. One such difference was the differential expression of miRNAs involved in glial lineage specification.

## Binding of transcription factors linked to DNA methylation and glial lineage specification are enriched at hypo-methylated loci in B-GICs

We used Homer (*Heinz et al., 2010*) to identify transcription factor (TF) binding sites that are enriched at hypo-methylated DMRs from B-GICs (17, 19, 30, and 31). We hypothesised that hypo-methylated DMRs were enriched for specific TF binding motifs linked to DNA methylation and/or glial lineage specification given the identification of miRNAs associated with glial differentiation (*Figure 1—figure supplement 1E*). Firstly, the hypo-methylated DMRs from the B-GICs-iNSC comparison were compared against DMRs from all GIC-iNSC comparisons. The top five enriched motifs were matches for an array of zinc-finger proteins, *HOX* genes, and families of factors such as hepatocyte nuclear factor 4 (*HNF4*), Kruppel-like factors (*KLF*), nuclear factor I (*NFI*), and distal-less homeobox (*DLX*) (*Figure 2A*). We noticed that some of these TFs, namely *NFI*, *ETV4*, *DLX*, and *KLF* have been previously linked to astrocyte/glial differentiation (*Sanosaka et al., 2017*; *Yin et al., 2015*; *Marshall and Goldman, 2002*).

To address the link to DNA methylation more stringently, motif enrichment analysis was performed on hypo-methylated DMRs identified from the B-GIC when compared to their syngeneic iNSC, relative to the hypo-methylated DMRs from nB-GICs when compared to their syngeneic iNSC. We reasoned that the hypo-methylated DMRs present in nB-GICs are a background of hypo-methylated DMRs. Therefore, comparing against this background should identify TFs that have potentially contributed to the hypo-methylation of DMRs specifically in the B-GICs. The top five enriched motifs were found to be matches for some of the same TFs identified in the first analysis such as *PLAGL1*, *NFI* family members, and ETS variant transcription factor 4 (*ETV4*) (*Figure 2B*). Interestingly, the fifth ranked enriched motif from this analysis was a strong match for members of the *SOX* family, known to be involved in cellular differentiation and neural development (*Kamachi and Kondoh, 2013*).

Next, Homer was used to identify the most enriched motifs in hyper-methylated DMRs from the B-GIC-iNSC comparisons, compared to DMRs from all other GIC-iNSC comparisons. We reasoned that hyper-methylated DMRs from B-GICs should not be enriched for any factors that may have caused the hypo-methylation bias. Therefore, any motifs found to be enriched in this comparison would have had to be disregarded. Importantly, *PLAGL1*, *NFI,* or *ETV4* were not identified in this analysis (*Figure 2C*).

Therefore, we hypothesised that TFs such as the *NFI* family could be responsible for the hypo-methylation bias, as they have been shown to demethylate specific loci (*Sanosaka et al., 2017*), including lineage defining genes.

To test this hypothesis, we leveraged the availability of patient-specific iNSCs to generate induced astrocyte progenitor cells (iAPCs). DNA methylome and transcriptome were analysed in five different progenitor lines at a differentiation stage, where cells co-expressed the astrocytic lineage marker *CD44* (*Liu et al., 2004*) and the neural progenitor marker *NESTIN* (*Bernal and Arranz, 2018*), in keeping with a astrocytic progenitor state (*Figure 2—figure supplement 1A, B*). Acquisition of a pro-inflammatory response upon IL6 treatment (*Tcw et al., 2017*), not observed in iNSC, and brisk proliferative activity (*Figure 2—figure supplement 1C, D*), confirmed the astrocytic commitment of the cells. PDGFRA[+] induced OPCs (iOPCs) were also generated for comparative analysis (*Figure 2—figure supplement 1E, F*). Inspection of the DNA methylation and RNAseq datasets obtained from these cells confirmed that iAPCs and iOPCs were epigenetically and transcriptionally distinct from one another and from the cells from which they were derived (*Figure 2D, E*). Furthermore, both cell types were found to be enriched for an astrocytic and oligodendrocyte signature, respectively (*Figure 2F, G, H*).

Differential methylation analysis was then carried out in a syngeneic fashion, between each of the five iAPCs and their matched iNSC samples. Hypo-methylated DMRs in all iAPC lines were identified and motif enrichment analysis performed on these regions, comparing against the sequences of all other iAPC - iNSC DMRs. The top five 'de novo' enriched motifs were matches for binding sites

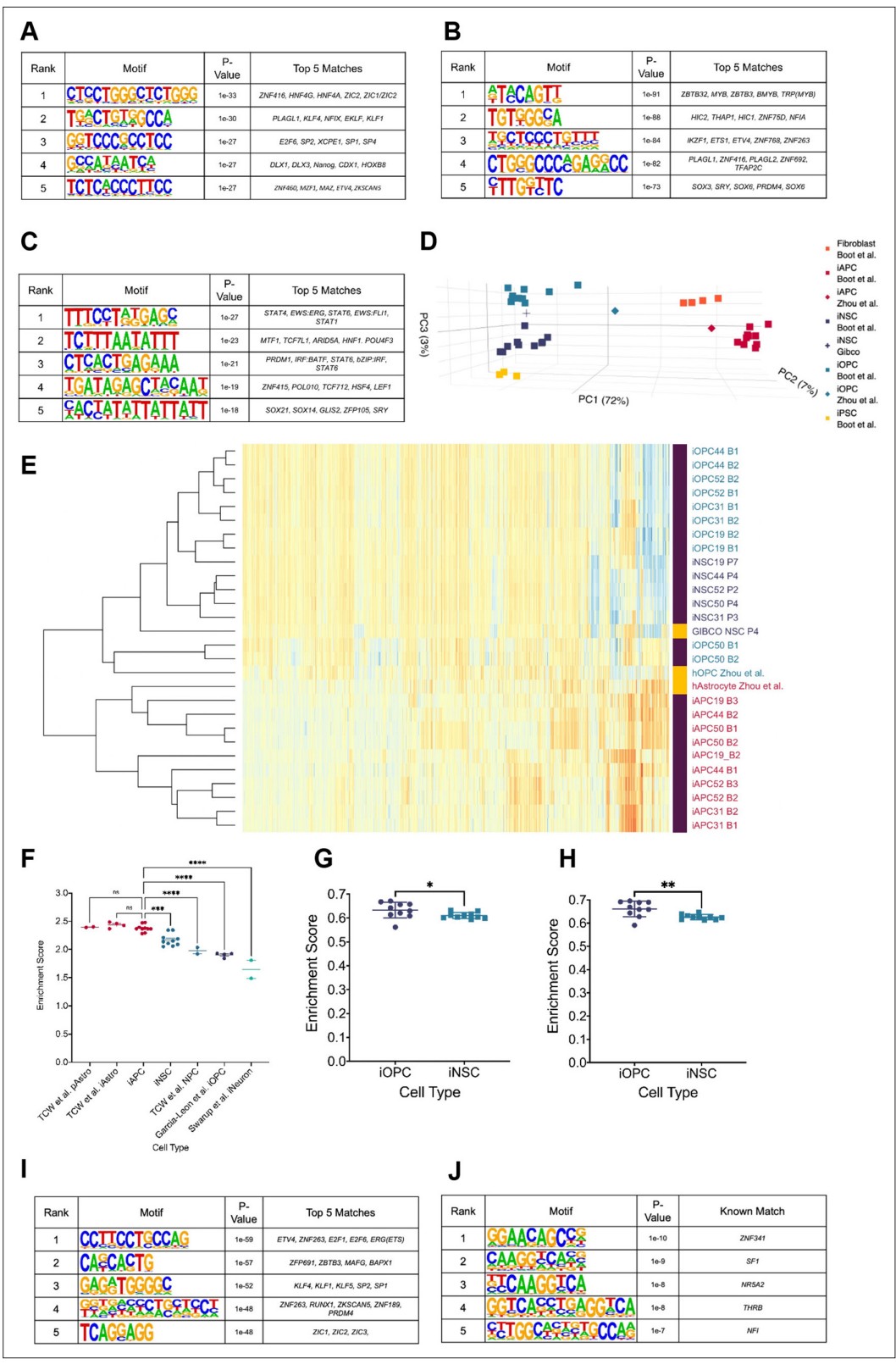

**Figure 2.** Hypo-methylated differentially methylated regions (DMRs) from bias-glioblastoma initiating cells (B-GICs) are enriched for transcription factor binding sites linked to glial differentiation. (**A**) Top five motifs enriched in all hypo-methylated DMRs from B-GICs as compared to all other DMRs from all GICs. (**B**) Top five motifs enriched in all hypo-methylated DMRs from B-GICs as compared to all the hypo-methylated DMRs from all other GICs.

*Figure 2 continued on next page*

*Figure 2 continued*

(**C**) Top five motifs enriched in all hyper-methylated DMRs from B-GICs as compared to all other DMRs from all GICs. (**D**) 3D principal component analysis (PCA) of methylation data from patient-derived fibroblasts, induced pluripotent stem cells (iPSCs), induced neural stem cells (iNSCs), induced astrocyte progenitor cells (iAPCs), induced oligodendrocyte progenitor cells (iOPCs), and publicly available reference datasets of NSCs, astrocytes, and oligodendrocyte precursor cells. (**E**) Unsupervised hierarchical clustering based on the top 5000 variable methylation probes of iAPCs, iNSCs, iOPCs (purple bar), and publicly available reference datasets (yellow bar). (**F**) Single sample gene set enrichment analysis (ssGSEA) enrichment scores for the astrocyte composite signature (ACS) of iAPCs (N=10), iNSCs (N=10), and publicly available reference datasets, statistical differences tested with one-way ANOVA. ssGSEA enrichment scores of iOPCs (N=10) and iNSCs (N=10) for the Oligodendrocyte Specific-300 (**G**) and Oligodendrocyte Enriched-300 (**H**) gene signatures, statistical significance tested using Mann-Whitney t-test. Top five de novo (**I**) and known (**J**) motifs enriched in all hypo-methylated DMRs in iAPCs, from each iAPC versus iNSC comparison.

The online version of this article includes the following source data and figure supplement(s) for figure 2:

**Source data 1.** Source data for *Figure 2*.

**Figure supplement 1.** Characterisation of induced astrocyte progenitor cell (iAPC) and induced oligodendrocyte progenitor cell (iOPC) obtained from induced neural stem cell (iNSC).

**Figure supplement 1—source data 1.** Source data for *Figure 2—figure supplement 1*.

previously found to be enriched in the comparison of hypo-methylated DMRs from B-GICs (*Figure 2A, B*). In particular, the top enriched motif was a strong match for *ETV4* confirming the link between this factor and hypo-methylated astrocyte associated loci (*Figure 2I*). The *NFI* binding motif was the fifth-ranked enriched motif among the '*known matches*' from Homer (*Figure 2J*). *ETV4* and *NFI* binding sites were not identified as enriched in either the 'de novo' or '*known matches*' when motif enrichment analysis was performed on hypo-methylated DMRs from iOPCs-iNSC comparison against all other DMRs (*Figure 2—figure supplement 1G, H*). Instead, hypo-methylated DMRs in iOPCs as compared to iNSCs were enriched for the binding sites of *ZIC* genes, *HOX* genes, *SOX* genes, and an array of zinc-finger proteins (*Figure 2—figure supplement 1G, H*).

In conclusion, hypo-methylated DMRs from B-GICs are enriched for TF binding sites, including *ETV4* and *NFI,* that have been linked to glial differentiation. iAPC hypo-methylated loci are similarly enriched for these TF binding sites, a phenomenon which is specific to the astrocytic lineage. These results raise the possibility that B-GICs have undergone glial priming prior to, or during, neoplastic transformation.

## A positive correlation between DNA hypo-methylation and astrocyte signature enrichment

Next, we assessed whether B-GICs were enriched for specific signatures of the glial lineage. We performed single sample gene set enrichment analysis (ssGSEA) on our cohort of GICs for an early radial glia (early-RG) signature (*Nowakowski et al., 2017*), a bespoke astrocytic signature, termed astrocyte composite signature (ACS) and two oligodendrocyte signatures (termed OPC Enriched-300 and OPC Specific-300), see Materials and methods for further details on these signatures. All GICs scored highly for the early-RG signature with little variability between lines (*Figure 3—figure supplement 1A*), likely reflecting the high degree of transcriptional similarity between GICs and NSCs. Four out of four B-GICs (17, 19, 30, and 31) had significantly higher enrichment scores for the ACS than the two OPC signatures (*Figure 3A*). GICs 50 and 26 showed a significant difference between the enrichment scores for the ACS and OPC signatures, however with a greater degree of variability between the two biological replicates. All other GICs showed no significant differences in enrichment scores for the gene signatures (*Figure 3A*). We further scored matching bulk FFPE tumour tissues for the ACS and two OPC signatures and found that the pattern of enrichment was not retained in the bulk tumours with very few tumours scoring highly for the ACS and the majority of tumour scoring highly for the OPC signatures (*Figure 3—figure supplement 1C*).

To validate these results, ssGSEA was performed on the GICs of the HGCC cohort (*Xie et al., 2015*). A subset of GICs that are highly enriched for the ACS were also identified in this cohort (*Figure 3—figure supplement 1B*). As there were no replicates for these GIC lines, we determined thresholds for the enrichment scores that separated the samples into enriched and non-enriched.

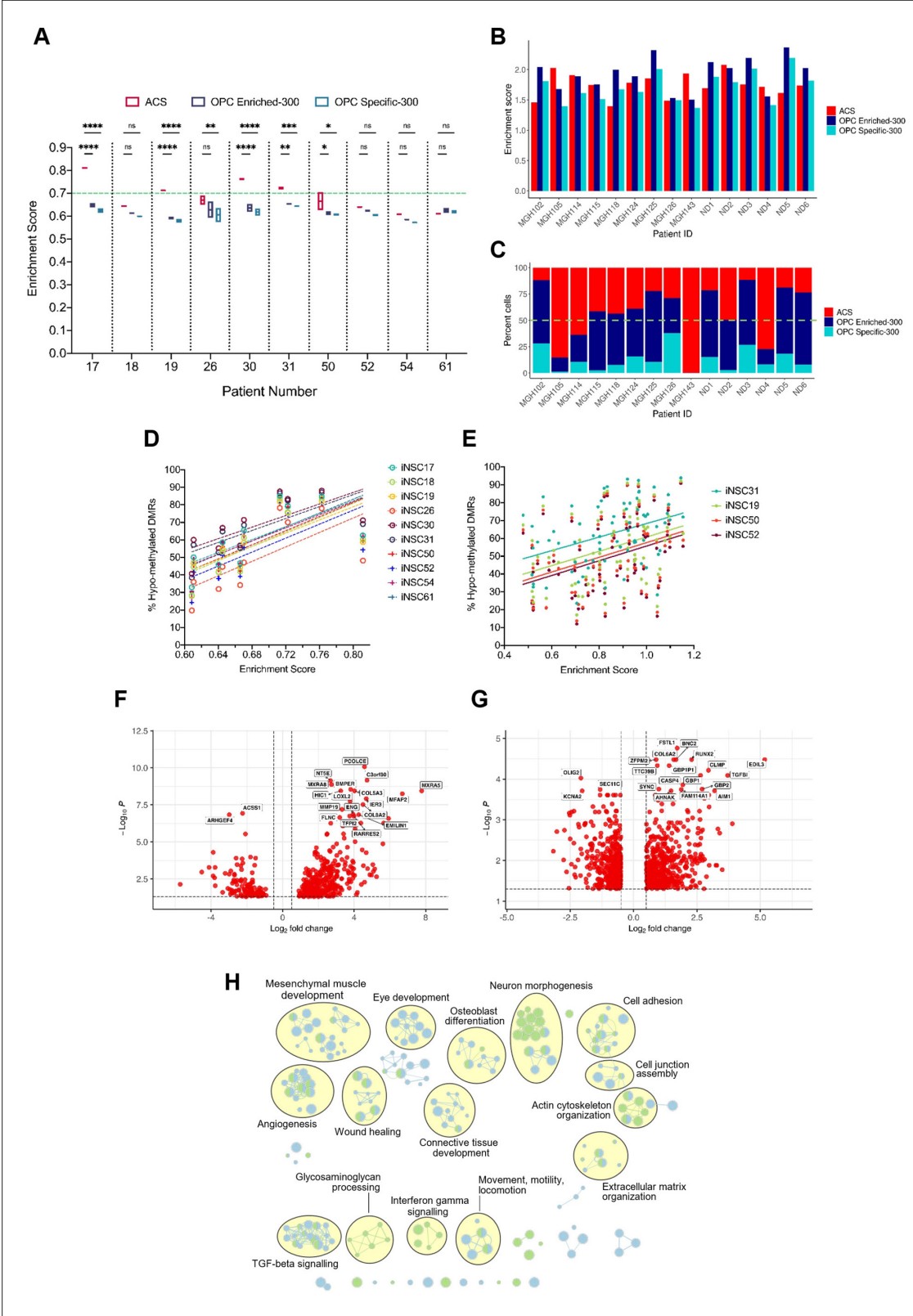

**Figure 3.** A positive correlation between DNA hypo-methylation and astrocyte signature enrichment. (**A**) Single sample gene set enrichment analysis (ssGSEA) enrichment scores of glioblastoma initiating cells (GICs) (N=2) for three different gene signatures; ACS (red), OPC Enriched-300 Signature (blue) and OPC Specific-300 Signature (teal), statistical significance tested using two-way ANOVA. Green dashed line indicating threshold for enrichment. (**B**) ssGSEA enrichment scores based on pseudo-bulk (aggregated within a patient) data of the cancer cell subset from the single-cell RNA

*Figure 3 continued on next page*

*Figure 3 continued*

sequencing (scRNAseq) glioblastoma tumour data from *Pombo Antunes et al., 2021*, and *Neftel et al., 2019*, for three different gene signatures, ACS (red), OPC Enriched-300 Signature (blue), and OPC Specific-300 Signature (teal). (**C**) Percentage of single cells for each tumour, which scored the highest for one of the three different signatures: ACS (red), OPC Enriched-300 Signature (blue), and OPC Specific-300 Signature (teal). Green dashed line indicating threshold for enrichment. (**D**) Scatter plot of the ACS enrichment score and the percentage of hypo-methylated DMRs for each of the GICs from our cohort, when comparing to iNSCs from our cohort. (**E**) Scatter plot of the ACS enrichment score and the percentage of hypo-methylated differentially methylated regions (DMRs) for each of the HGCC GICs, when comparing to induced neural stem cells (iNSCs) from our cohort. (**F**) Volcano plot of differentially expressed genes (DEGs) identified from the comparison of BE-GICs versus nBE-GICs (glm model used for differentially expressed [DE] analysis). (**G**) Volcano plot of DEGs identified from the comparison of BE-GICs versus nBE-GICs from the HGCC cohort (glm model used for DE analysis). (**H**) Summary of pathway analysis performed using gProfiler, using the DEGs identified in (**F**) and (**G**).

The online version of this article includes the following source data and figure supplement(s) for figure 3:

**Source data 1.** Source data for *Figure 3*.

**Figure supplement 1.** Characterisation of the positive correlation between DNA hypo-methylation and astrocyte signature enrichment.

**Figure supplement 1—source data 1.** Source data for *Figure 3—figure supplement 1*.

Firstly, enriched cell lines had to have a higher-than-average enrichment score for the ACS (>0.8429). Secondly, ACS enrichment scores had to be 10% greater than any of the OPC signature scores. Given these criteria, 55% of the samples were found to be enriched for the ACS in the HGCC dataset, compared to approximately 64% in our cohort, hence confirming an ACS enrichment in an independent GIC cohort.

Next, single-cell RNA sequencing (scRNAseq) data of glioblastomas from two different sources (*Neftel et al., 2019*; *Pombo Antunes et al., 2021*) were interrogated to validate the signature enrichment observed in a proportion of GICs. The analysis of the integrated scRNAseq dataset of 15 glioblastomas showed presence of cancer cells, immune cells, oligodendrocytes, and a small cluster of endothelial cells. To identify patients enriched for the ACS signature, we employed two strategies. Firstly, by treating all the single cancer cells' expression as pseudo-bulk tumour tissue, we were able to identify a subset of tumours, which scored higher for the ACS than the two OPC signatures (MGH105, MGH114, MGH143, ND2, and ND4) (*Figure 3B*). Secondly, we determined enrichment scores for single cancer cells from each tumour and then assigned a signature identity to each cell to quantify the percentage of ACS, OPC Enriched-300, or OPC Specific-300 cells per tumour. Encouragingly the same tumours found to have a higher ACS enrichment score based on the pseudo-bulk analysis, had increased proportion of ACS cells: 75.07% ± 19.40% compared to 26.26% ± 11.69% in the remaining tumours (*Figure 3C*). Finally, we applied these same two approaches but using the six cancer gene signatures from *Neftel et al., 2019* (astrocytic: 'AC'; mesenchymal: 'MES1' and 'MES2'; oligodendrocyte progenitor-specific: 'OPC'; and neural progenitor-specific: 'NPC1' and 'NPC2'). The same five patients (MGH105, MGH114, MGH143, ND2, and ND4) had higher enrichment scores and proportions of cancer cells, assigned to the astrocytic signature ('AC') compared to the OPC signature, and the same held true for MGH115, MGH118, MGH124, MGH125, ND1, and ND6. For patients MGH143, ND2, and ND4, as well as for MGH115, MGH118, and ND5, the 'AC' signature showed highest enrichment and percent assigned cells within the six studied gene modules (*Figure 3—figure supplement 1D, E*). Interestingly, we noted that the ACS from our study enveloped the 'MES1', 'MES2', and 'AC' signatures from *Neftel et al., 2019* (*Figure 3—figure supplement 1F, G*) but scoring B-GICs from our cohort for the *Neftel et al., 2019*, signatures did not allow to further dissect the enrichment type as three out of four B-GICs (17, 19, and 30) scored most highly for the 'MES1' and 'MES2' signatures as well as the 'AC' signature, GIC31 scored very similar for all Neftel et al. signatures (*Neftel et al., 2019*; *Figure 3—figure supplement 1H*).

A significant positive correlation was found between GIC ACS signature enrichment scores and the percentage of hypo-methylated DMRs in both our (*Figure 3D, L*) and in the HGCC cohorts (*Figure 3E, M*), raising the possibility that the two findings could be causally related.

To further explore the link between hypo-methylation bias and ACS enrichment, a differential expression analysis comparing B-GICs enriched for the ACS (termed bias-enriched [BE-GICs]), and GICs not harbouring a hypo-methylation bias and not enriched for the ACS (termed non-bias-enriched [nBE-GICs]) was performed. The BE-GIC group contained GICs 17, 19, 30, and 31, whilst the nBE-GIC group contained GICs 18, 26, 50, 52, 54, and 61. We reasoned that DEGs from this comparison could highlight key biological differences between BE-GICs and nBE-GICs. Of the 465 DEGs

(*Figure 3F*), 47 were part of the ACS (*Figure 3—figure supplement 1I*) (Fisher's exact test, p-value < 0.00001). The same differential expression comparison in the HGCC cohort identified a total of 902 DEGs (*Figure 3G*), of which a significant number (52) were also present in the ACS (*Figure 3—figure supplement 1J*) (Fisher's exact test, p-value < 0.00001). When considering the top DEGs between the two groups in the HGCC cohort, *RUNX2* was up-regulated whilst *OLIG2* was down-regulated in BE-GICs as compared to the nBE-GICs. This is an interesting finding, as *RUNX2* has been shown to drive astrocytic differentiation, and *OLIG2* is a well-known master regulator of oligodendroglia differentiation (*Tiwari et al., 2018*). An overlap of 105 genes from the 465 DEGs identified in our cohort and the 902 DEGs identified in the HGCC cohort was found (*Figure 3—figure supplement 1K*) (Fisher's exact test, p-value < 0.00001). Finally, pathway analysis on the two lists of DEGs revealed a shared enrichment for pathways associated with the extracellular matrix, morphogenesis, cell adhesion, angiogenesis, locomotion, wound healing, and cytokine signalling (*Figure 3H*), suggesting a deregulation of these pathways in BE-GICs.

In conclusion, we show in two independent GIC cohorts a positive correlation between the extent of hypo-methylation bias and the ACS enrichment score, in keeping with B-GICs being enriched for an astrocytic gene signature. An enrichment for an astrocytic signature is confirmed also in a proportion of glioblastoma at single cell level. Furthermore, when comparing BE-GICs to nBE-GICs, a significant number of DEGs that are present in the ACS are found and a predicted impact on cell movement/invasion is identified.

## Increased invasion in xenografts derived from BE-GICs and role of SRPX2 in regulating invasion in vitro

Next, we set out to assess whether BE-GICs would give rise to tumours with distinct invasive properties as compared to those generated from other GICs. Xenografts derived from our GIC lines (*Vinel et al., 2021*) were stained for human vimentin on three levels. QuPath (*Bankhead et al., 2017*), a machine learning-based pixel classifier, was used for tissue detection from glass, vimentin staining detection, and tumour core detection. We calculated the invasiveness index (II) for each xenograft, defined as the ratio of area covered by infiltrating tumour cells to the area of the tumour core (gross tumour area/tumour core area) independent of tumour size (*Amodeo et al., 2017*). Xenografts derived from BE-GICs showed increased invasiveness (*Figure 4A, B, C*), despite having a smaller tumour core (*Figure 4—figure supplement 1B, C, D*).

To identify genes that may contribute to the phenotypic characteristics of the BE-GICs, the following differential expression comparisons were performed on a patient-by-patient basis, for four patients whereby two biological replicates of iAPCs, iNSCs, and GIC were available: GIC versus iAPC, iAPC versus iNSC, and GIC versus iNSC. We aimed to identify genes that play a role in the differentiation of iAPCs that are also deregulated in GICs and may be responsible for astrocytic signature enrichment in BE-GICs. Initially, DEGs present in both the GIC versus iNSC, and iAPC versus iNSC comparisons were selected for further analysis, for three reasons. Firstly, these DEGs could play a role in the differentiation of iNSCs to iAPCs if they are differentially expressed between these two cell types. Secondly, as these DEGs are also differentially expressed between GICs and iNSCs, they may play a role in neoplastic transformation. Thirdly, as these DEGs are not differentially expressed between iAPCs and GICs, they could contribute to the enrichment for an astrocytic signature. Once DEGs meeting these criteria had been identified across our four patients, we further selected for genes also present in the ACS, and specifically identified in the comparison of BE-GIC 19 and 31 and shared between the two (summarised schematically in *Figure 4—figure supplement 1A*).

GLI pathogenesis related 1 (*GLIPR1*), sushi repeat containing protein X-linked 2 (*SRPX2*), and leukaemia inhibitory factor (*LIF*) were identified (*Figure 4D*). *GLIPR1* is very well studied in cancer (*Awasthi et al., 2013*) and has been found to regulate migration and invasion of glioma cells (*Ziv-Av et al., 2015*; *Giladi et al., 2015*). Similarly, *LIF* is well studied in glioma and has been shown to contribute to the maintenance of glioma-initiating cell self-renewal (*Edwards et al., 2017*; *Peñuelas et al., 2009*). Moreover, *LIF* has been shown to help mediate astrocyte differentiation (*Nakashima et al., 1999*). The pathogenic role and impact of *SRPX2* in GBM (*Tang et al., 2016*) is less well characterised, despite been well studied in cancer (*Hong et al., 2018*; *Tanaka et al., 2009*; *Lin et al., 2017*; *Zhang et al., 2018*). Analysis of the pattern of expression of *SRPX2* in our dataset found that it was up-regulated in the comparisons between iNSC and iAPC across all patients, and its expression was

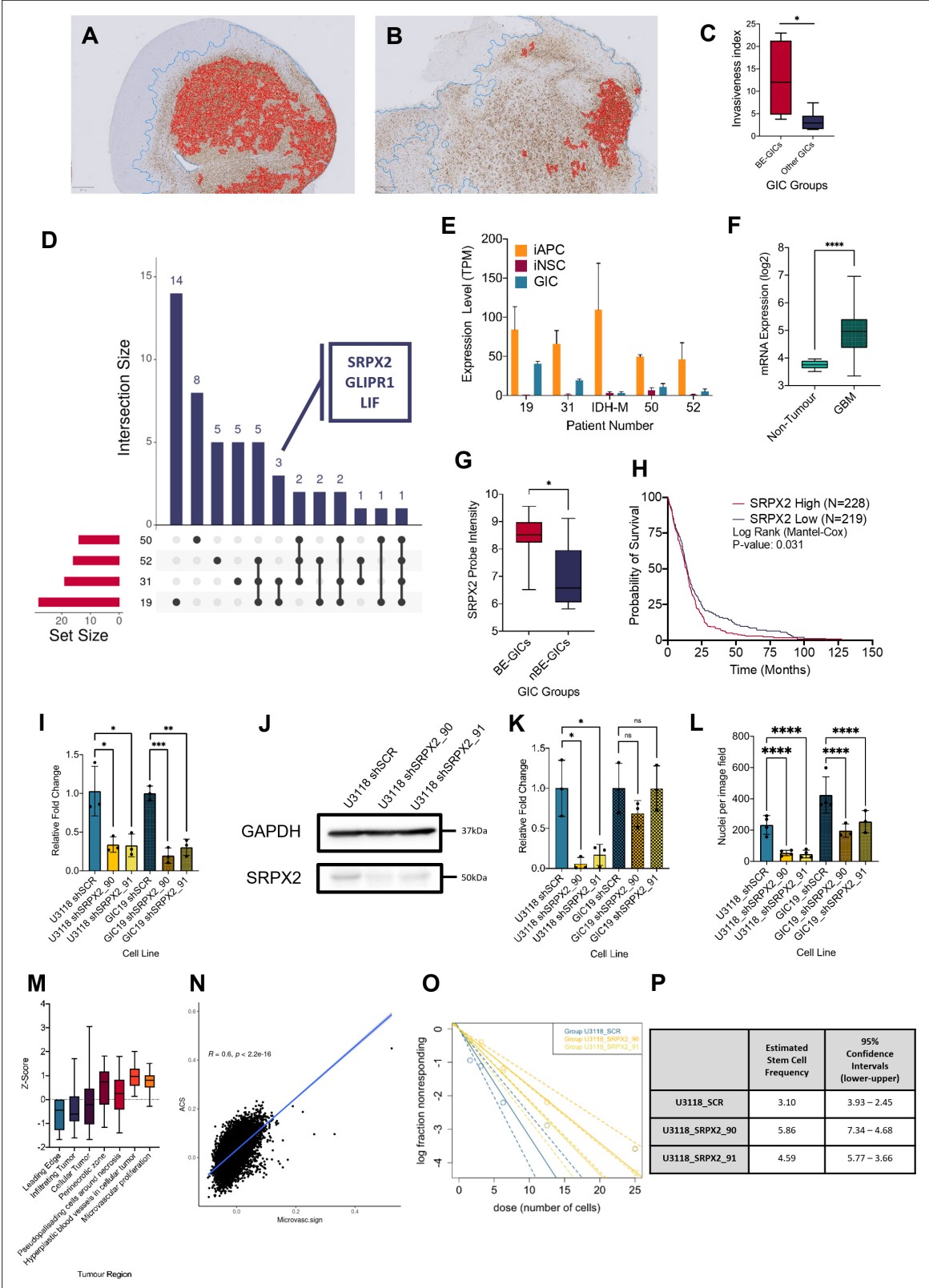

**Figure 4.** Increased invasion in xenografts derived from BE-GICs and role of SRPX2 in regulating invasion in vitro. (**A**) and (**B**) Representative images of human vimentin-stained xenograft tumours with overlayed image analysis. Red outline is the detected tumour core, blue outline is the detected gross tumour edge. (**C**) Invasive index scores of xenografts from BE-GICs (N=4) and other GICs (N=8), statistical significance tested using un-paired t-test. (**D**) Overview of significant differentially expressed genes (DEGs) that are present in the astrocyte composite signature (ACS) and are identified

*Figure 4 continued on next page*

*Figure 4 continued*

in glioblastoma initiating cell (GIC) versus induced neural stem cell (iNSC) and induced astrocyte progenitor cell (iAPC) versus iNSC comparisons across patient comparisons. (**E**) Expression (transcripts per million [TPM]) of SRPX2 in the three cell types analysed: iAPC (orange), iNSC (red), and GIC (turquoise). (**F**) Expression of target genes in glioblastoma tissue (N=528) as compared to non-tumour tissue (N=10), data acquired from Gliovis (*Bowman et al., 2017*), statistical significance tested using Mann-Whitney t-test. (**G**) Expression of target genes in BE-GICs (N=13) versus nBE-GICs (N=6) from the HGCC cohort, statistical significance tested using Mann-Whitney t-test. (**H**) Kaplan-Meier curve for GBM patients with high expression (red) versus low expression (blue) of SRPX2, produced using TCGA data available on Gliovis (*Bowman et al., 2017*). (**I**) Relative fold change in mRNA expression of SRPX2 as determined by qPCR for GIC short hairpin RNA (shRNA) knockdown lines, statistical significance tested using t-test (N=3). (**J**) Representative western blot of SRPX2 in U3118 shRNA knockdown lines. (**K**) Relative fold change in SRPX2 protein expression as determined by western blot for GIC shRNA knockdown lines, statistical significance tested using t-test (N=3). (**L**) Invasion assay results: average number of nuclei per image field of GIC SRPX2 knockdown lines, statistical significance tested using two-way ANOVA (N=3-4). (**M**) Z-score of SRPX2 expression in different tumour regions according to Ivy GAP (*Puchalski et al., 2018*) (N=19-111). (**N**) Correlation of ACS enrichment scores (y-axis) and enrichment scores for a signature of the top 200 up-regulated genes in regions of microvascular proliferation and hyperplastic blood vessels (x-axis) in single tumour cells from *Pombo Antunes et al., 2021*. (**O**) Neurosphere assay results: log fraction of the number of non-responding cultures at specified cell counts for U3118 SRPX2 knockdown lines. (**P**) Table of estimated stem cell frequencies and confidence intervals as determined by the neurosphere assay results and extreme limiting dilution assay analysis.

The online version of this article includes the following source data and figure supplement(s) for figure 4:

**Source data 1.** Source data for *Figure 4*.

**Figure supplement 1.** Characterisation of xenografts derived from BE-GICs and role of SRPX2 in regulating glioblastoma initiating cell (GIC) properties.

**Figure supplement 1—source data 1.** Source data for *Figure 4—figure supplement 1*.

specifically higher in the GICs of patients 19 and 31 (*Figure 4E*), as expected. Given our limited cohort size, the deregulation of SRPX2 was further validated in other datasets and found that it was more highly expressed in glioblastoma tumour tissue as compared to non-tumour brain tissue according to TCGA data (*Figure 4F*) and in BE-GICs from the HGCC cohort (*Figure 4G*). Using survival data from the TCGA we also found that high SRPX2 expression confers a worse prognosis (*Figure 4H*). To elucidate the role of *SRPX2*, the HGCC BE-GIC line, U3118, and GIC19 were transduced with lentiviral vectors, carrying short hairpin RNA (shRNA) constructs targeting *SRPX2* (termed SRPX2_90 and SRPX2_91) and a scramble control. The mRNA levels of *SRPX2* were decreased upon silencing (*Figure 4I*) and western blotting confirmed effective knockdown of the gene at the protein level (*Figure 4E, J, K*).

The differential expression comparison between BE-GICs and nBE-GICs identified cell movement/ invasion among the differentially enriched pathways and BE-GICs gave rise to more invasive tumours upon intracerebral injection in mice. Cell movement/invasion is a process that is also impacted in astrocytic progenitors, which are more motile as compared to NSC (*Schiweck et al., 2018*). For these reasons, the impact of SRPX2 knockdown on the invasive phenotype of GICs was assessed using a transwell invasion assay. The nuclei of cells that moved across the transwell membrane were fixed 24 hr after seeding, then stained and counted as described in the Materials and methods. On average, a statistically significant lower number of nuclei per image field were found in both GIC19 and U3118 *SRPX2* knockdown lines (*Figure 4L*) (p-value < 0.0001).

Interestingly, interrogation of the Ivy GAP (*Puchalski et al., 2018*) resource revealed *SRPX2* expression predominantly in perivascular regions, including areas of microvascular proliferation and hyperplastic blood vessels, whilst it was down-regulated in the parenchymal leading edge and infiltrating tumour (*Figure 4M*). Given that gliomas invade mainly along two routes, the white matter tracts and the blood vessels (*Giese and Westphal, 1996*), it could be speculated that *SRPX2* may play a role in the latter and not the former in the subset of glioblastoma with hypomethylation bias/astrocytic signature enrichment. Although functional studies in vivo will be required to substantiate this hypothesis, we scored single tumour cells from the scRNAseq data of *Pombo Antunes et al., 2021*, for both the peri/microvascular signature (top 200 genes up-regulated in these regions according to Ivy GAP; *Puchalski et al., 2018*), the ACS and the two OPC signatures. This showed that the scores for the peri/ microvascular signature and the ACS were positively correlated (*Figure 4N*), whilst the former and the OPC signatures were not (*Figure 4—figure supplement 1F, G*).

Proliferation was not affected by *SRPX2* knockdown in GICs (*Figure 4—figure supplement 1H*). Variable results were obtained when self-renewal capacity was assessed by means of neurosphere extreme limiting dilution assay. Less neurospheres formed in the U3118 shRNA *SRPX2* knockdown line

at lower cell counts (6.25, 3.125, and 1.5625 cells per well) (*Figure 4O, P*), but this was not reproduced in the GIC19 shRNA *SRPX2* knockdown lines (*Figure 4—figure supplement 1I, J*).

In conclusion, we found that BE-GICs gave rise to more invasive tumours in a xenograft model and that genes involved in migration/invasion are deregulated in these cells. Among these, *SRPX2* plays a role in invasive properties of BE-GICs, raising the possibility that *SRPX2* could be further explored as a potential therapeutic target for glioblastoma with a hypo-methylation bias and ACS enrichment.

## Glioblastoma enriched for an astrocytic signature display an altered immune landscape

Pathway enrichment analysis performed on the lists of DEGs generated from the comparisons of BE-GICs versus nBE-GICs identified a deregulation of immune-related pathways – TGF-beta signalling and interferon-gamma signalling (*Figure 3H*), raising the possibility of differences in the tumour microenvironment of these glioblastomas. Among the 47 DEGs shared with the ACS identified when comparing BE-GICs and nBE-GICs (*Figure 3—figure supplement 1I*) was retinoic acid receptor responder 2 (RARRES2) – a chemoattractant, which binds to Chemerin chemokine-like receptor 1 (CMKLR1). *RARRES2* was found to be significantly up-regulated in BE-GICs compared to nBE-GICs, in both our cohort and the HGCC cohort, and in non-G-CIMP tumour tissue relative to non-tumour brain tissue (*Figure 5A and B* and *Figure 5—figure supplement 1A*). Moreover, high expression of *RARRES2* is correlated with a worse prognosis (*Figure 5—figure supplement 1B*). Significantly, *RARRES2* is well known to play a role in inflammation (*Mattern et al., 2014*) and promoting the migration of plasmacytoid DC, macrophages, and NK-cells (*Treeck et al., 2019*). In scRNAseq data from *Pombo Antunes et al., 2021*, and *Neftel et al., 2019*, the receptor CMKLR1 is expressed in tumour-associated macrophages (TAMs) from newly diagnosed GBMs (*Figure 5C, D*), particularly in non-hypoxic TAMs (such as SEPP1-hi TAMs) as compared to hypoxic TAMs in recurrent glioblastomas (*Figure 5E, F*).

Therefore, we leveraged these datasets to analyse the immune composition of tumours with an enrichment for an astrocytic signature or increased proportion of astrocyte-like cells (*Figure 3B, C*). Both the myeloid and the lymphocyte clusters of the integrated scRNAseq data were selected and re-clustered, yielding various subpopulations of TAMs, dendritic cells (DC), mast cells, B, T, NKT, and NK cells (*Supplementary file 2*, *Supplementary file 3*). The ACS/OPC pseudo-bulk score ratio had a significant positive correlation with the proportions of TAMs (in particular of SEPP1-hi moTAMs and cDC2) and of proliferating CD8 T cells (*Figure 5*). Conversely, a significant negative correlation was found between the proportion of cancer cells and the ACS/OPC pseudo-bulk score ratio (*Figure 5I*). Similar trends were observed when, utilising the single-cell resolution of the scRNAseq data, we correlated the proportion of TAMs, proliferating CD8 T cells, or cancer cells against the proportion of ACS-like tumour cells per patient (cells with higher ACS signature, compared to the two OPC signatures) (*Figure 5J-L*). The comparison of the proportion of ACS-like cancer cells against the fractions of monocytes and IFN-response CD8 T cells per patient also yielded a significant positive correlation. Furthermore, we show that these results were reproduced, when the signatures from *Neftel et al., 2019*, were used to determine the proportions of astrocyte-like cancer cells per patient and the latter were correlated to the proportions of immune and cancer cells in each tumour (*Figure 5M-O*). Staining for CD68 in the FFPE tumour tissue from our cohort found a trend for an increased percentage of CD68[+] cells in tumours of BE-GICs relative to nBE-GIC tumours (*Figure 5—figure supplement 1C*). Finally, the opposite trends were observed when we correlated the proportion of TAMs, proliferating CD8 T cells, or cancer cells against the proportion of OPC and NPC-like cancer cells (as determined by *Neftel et al., 2019*, signatures), however not all of the findings were statistically significant (*Figure 5—figure supplement 1D-I*).

In conclusion, a significant up-regulation of *RARRES2* is identified in BE-GICs, and an increased proportion of TAMs and CD8 proliferative T cells is found in tumours with a larger fraction of astrocyte-like tumour cells. This may indicate that the composition of the TME is different in tumours with an astrocytic signature enrichment.

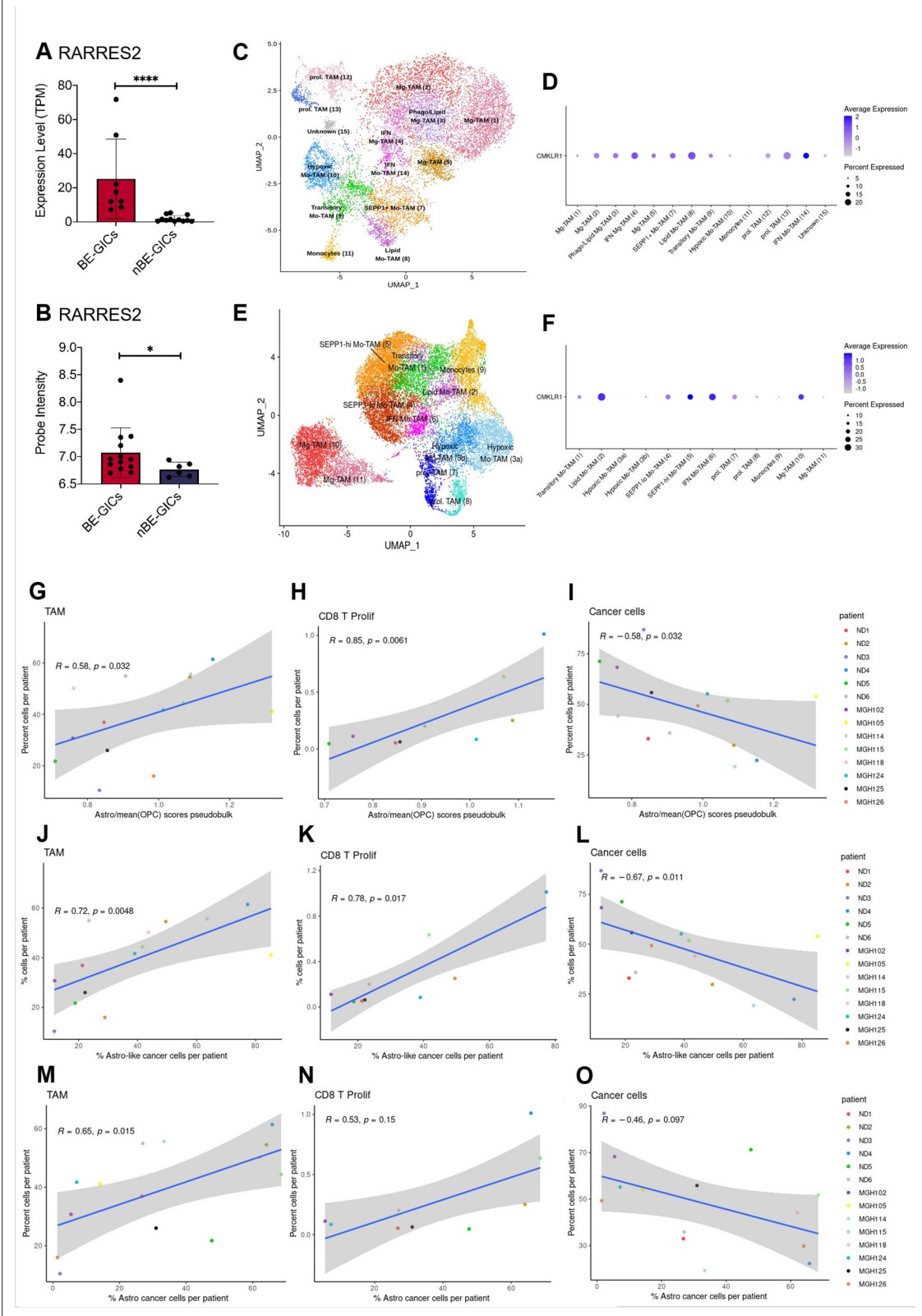

**Figure 5.** GBM enriched for an astrocytic signature display an altered immune landscape. (**A**) Expression of RARRES2 in BE-GICs (N=8) and nBE-GICs (N=12) in our cohort, determined by RNAseq. (**B**) Expression of RARRES2 in BE-GICs (N=13) and nBE-GICs (N=6) in the HGCC cohort. (**C**) UMAP of monocyte and tumour-associated macrophage (TAM) cell clusters from the Antunes et al. (*Tarazona et al., 2015*) newly diagnosed glioblastoma tumour data; the cells are coloured by cell type. (**D**) Expression of CMKLR1 across different immune cell type clusters from panel (**C**). (**E**) UMAP of monocyte and

*Figure 5 continued on next page*

*Figure 5 continued*

TAM clusters from the Antunes et al. (*Tarazona et al., 2015*), recurrent glioblastoma tumour data; the cells are coloured by TAM subtype. (**F**) Expression of CMKLR1 across different TAM subtype clusters from panel (**E**). Scatter plot, comparing the ratio of the astrocyte composite signature (ACS) and the mean OPC pseudo-bulk enrichment scores, and the proportion of TAM cells (**G**), CD8 proliferative T cells (**H**), and cancer cells (**I**) from the same tumour. Scatter plot, comparing the proportion of ACS-enriched cancer cells and the proportion of TAM cells (**J**), CD8 proliferative T cells (**K**), and cancer cells (**L**) from the same tumour, corresponding to the dataset from (**G–I**). Scatter plot, comparing the proportion of AC-enriched and the proportion of TAM cells (**M**), CD8 proliferative T cells (**N**), and cancer cells (**O**) from the same tumour, corresponding to the dataset from (**G–I**). Spearman's rank correlation coefficient and the corresponding p-value are noted on each scatter plot. The blue lines represent smoothed conditional means using general linear model, while the grey areas on the plots denote the confidence interval around the smooth (using the geom_smooth function of ggplot2).

The online version of this article includes the following source data and figure supplement(s) for figure 5:

**Source data 1.** Source data for *Figure 5*.

**Figure supplement 1.** scRNA analysis of the TME composition in glioblastoma enriched for an astrocytic signature.

**Figure supplement 1—source data 1.** Source data for *Figure 5—figure supplement 1*.

## Discussion

We have identified a novel DNA hypo-methylation bias in a proportion of glioblastoma, which is ontogenetically linked to astrocyte progenitors. Hypo-methylated loci are enriched for binding motifs of TFs known to be involved in astrocyte differentiation, and DEGs and miRNAs known to play a role in astrocyte lineage commitment and differentiation were detected in B-GICs. At a functional level, BE-GICs are characterised by increased invasion in vivo, enhanced *SRPX2*-regulated invasive properties in vitro, and an altered immune microenvironment.

Alterations of the DNA methylome have been previously described in genetically defined high-grade gliomas. In particular, the G-CIMP hyper-methylator phenotype characterises IDH-mutant gliomas (*Tahiliani et al., 2009*), where mutated IDH gains the ability to convert α-KG to 2-HG, which functions as an oncometabolite whilst also being a competitive inhibitor of α-KG-dependent dioxygenases such as TET enzymes (*Tahiliani et al., 2009*). Consequently, IDH-mutant glioblastoma cells accumulate DNA methylation as TET enzymes are inhibited from removing methylation marks. A hypo-methylation bias has been previously found also in the paediatric H3 G34 mutant glioma subgroup (*Sturm, 2012*), where mutations in the histone H3 variant H3.3 (H3F3A) block *SETD2* binding, leading to loss of H3K36 methylation, which in turn is linked to DNA methylation (*Rose and Klose, 2014*). Alternatively, mutation and loss of *ATRX*, which is consistently found in G34 mutant glioblastoma, could contribute to hypo-methylation at highly repeated sequences such as rDNA (*Gibbons et al., 2000*). The DNA hypo-methylation bias described here is found in a proportion of IDH-wildtype glioblastomas, which are enriched for an astrocytic signature. The pan-cancer analysis of DNA methylomes, published in 2018 (*Saghafinia et al., 2018*), reported that a proportion of IDH-wildtype gliomas exhibited high percentages of hypo-methylated loci, which correlated with a stemness signature (*Malta et al., 2018*), based on hypo-methylation of specific loci enriched for the *SOX2-OCT4* binding motif. We did not find enrichment for the *SOX2-OCT4* motif in the hypo-methylated DMRs from B-GICs, but rather an enrichment for TF binding sites known to play a role in glial/astrocyte differentiation, which is consistent with the enrichment for an astrocytic signature. Importantly, we have shown that the hypo-methylation bias is not induced in GIC by in vitro culture as it was confirmed in the respective tumour tissue. Likewise, we have taken advantage of publicly available single-cell transcriptome datasets to confirm that enrichment for our bespoke astrocytic signature as well as published astroglia signatures (*Neftel et al., 2019*) is found in a proportion of GBM.

Of particular interest was the finding of an enrichment for members of the *NFI* family in the hypo-methylated DMRs from B-GICs as previous studies have shown that astrocyte differentiation requires Nfia-induced demethylation of key astrocyte lineage specifying genes (*Sanosaka et al., 2017*; *Namihira et al., 2009*). In fact, *Sanosaka et al., 2017*, showed that the methylome underpins the differentiation potential of NSCs rather than gene expression itself. They found that E11.5 embryonic mouse NSCs were lineage-restricted and only giving rise to neurons, whilst E18.5 NSCs had an increased proportion of hypo-methylated loci and were multipotent – being able to give rise to glial and neuronal lineages. Furthermore, this study also showed that DMRs with reduced methylation (hypo-methylated) in E18.5 NSCs as compared to E11.5 NSCs are enriched for *NFI* binding sites, and that this gene family was responsible for the loss of DNA methylation and gain of multipotency for

the glial lineage. It is intriguing that these previous studies may provide an interpretative framework as to why we find a positive correlation between the hypo-methylation bias and ACS enrichment. It is conceivable that B-GICs may arise from a neural progenitor population that has undergone priming for glial differentiation, for example by the *NFI* TF family. Other TF binding sites we found enriched in hypo-methylated loci are also known to be involved in glial/astrocyte differentiation, such as *ETV4* (*Ghosh et al., 2016*; *Tiwari et al., 2018*) and *PLAGL1*, the latter by transactivating *Socs3*, a potent inhibitor of pro-differentiative *Jak/Stat3* signalling, thereby preventing precocious astroglia differentiation (*Schmidt-Edelkraut et al., 2013*). We found a significant enrichment for genes of the ACS in the DEGs between B-GICs and nB-GICs in both our and the HGCC datasets. Likewise, the five differentially expressed miRNAs identified in this comparison – miR-4443, miR-1275, miR-196, miR-5100, and miR-1268, which have been previously linked to cancer or glioblastoma pathogenesis (*Mai et al., 2019*; *Gao et al., 2019*; *Ma et al., 2012*; *Yang and Wang, 2016*; *Wang et al., 2017b*), have also been linked either directly or indirectly to glial differentiation. These include miR-196 known for its regulatory role of Homeobox (HOX) genes (*Sehm et al., 2009*) in both a healthy and malignant developmental context and miR-1275 previously shown to be involved in glial lineage specification (*Mai et al., 2019*; *Katsushima et al., 2012*). Taken together, the results of the binding motif analysis and of the differential expression analysis raise the possibility that BE-GICs arise from a neural progenitor, which has undergone glial/astrocyte priming. Overexpression of *NFI* family members in nBE-GICs will be required to assess whether changes in DNA methylation and gene signature enrichment can be elicited, or whether this enrichment is a consequence of the hypo-methylation bias. Alternatively, disruption of the *NFI* binding motifs across the genome, for example by means of CRISPR-Cas9 system (*Ran et al., 2013*) could be carried out to assess the effect on DNA hypo-methylation and gene signature enrichment.

We noted that whilst the hypo-methylation bias was retained in bulk FFPE tumour tissue, the ACS enrichment was not. It is possible that this is due to the lower RNA quality, as compared to excellent DNA quality obtained from FFPE material, which limit the transcriptome characterisation. It is also possible that the transcriptome is more dynamic and prone to change, and so for example whilst GICs pass on their epigenetic signature to daughter cells when they proliferate, the regulation of their transcriptome may not be equally inherited by the daughter cells. Finally, the cellular heterogeneity, including the tumour microenvironment, could also be obscuring the astrocyte signature enrichment in the parental tumour.

Importantly, when comparing DEGs between BE-GICs and nBE-GICs, a predicted impact on cell movement/invasion was identified. In keeping with this prediction, xenografts generated from BE-GICs were found to grow more invasively as compared to those generated by nBE-GICs. This is of particular interest in glioblastoma, given the diffusely infiltrative growth of these tumours, which plays a significant role in limiting the effectiveness of the current therapies. Among the genes deregulated, *SRPX2* stood out as it has been previously shown to be associated with poor prognosis, and to promote tumour progression and metastasis in primary GICs (*Tang et al., 2016*). Indeed, we show that silencing of this gene leads to impaired invasion in two BE-GIC lines in in vitro assays, although in vivo validation will be required to substantiate this finding.

The immune microenvironment plays a crucial role in tumour pathogenesis, including glioblastoma (*Pombo Antunes et al., 2021*; *Dumas et al., 2020*). We have identified a significant deregulation of genes involved in immunomodulatory pathways in BE-GICs and analysis of scRNAseq datasets of glioblastoma has confirmed an altered immune landscape in glioblastoma with an ACS signature enrichment. The significant correlation between the ACS signature enrichment and the number of TAMs is of particular interest given the up-regulation of *RARRES2* in BE-GICs and the role of TAMs in glioblastoma invasion (*Andersen et al., 2021*). Interestingly, a SEPP1-hi phenotype was observed in these TAMs corresponding to an anti-inflammatory phenotype (*Pombo Antunes et al., 2021*), which raises the possibility that they could play a pro-tumourigenic role in these glioblastomas.

A recent methylome analysis of various tumour types has found that global methylation loss correlates with increased resistance to immunotherapy and immune evasion signatures (*Jung et al., 2019*), hence the identification of GBMs with these features could have important implications in patient stratification for immunomodulatory treatments.

# Materials and methods

## Tissue culture

We have previously described a novel experimental pipeline, SYNGN, to derive GICs and EPSCs as well as iNSCs from patients who underwent surgical resection of glioblastoma (*Vinel et al., 2021*). The use of human tissue samples was approved by the National Research Ethics Service (NRES), University College London Hospitals NRES Project ref 08/0077 (S Brandner); Amendment 1 17/10/2014. In brief, at the time of operation, tumour tissue and a thin strip of the dura mater were obtained. Fibroblasts were isolated from the dura mater, propagated, and reprogrammed to generate EPSCs, which were further differentiated into iNSCs, iAPCs, and iOPCs. All media recipes can be found in *Supplementary file 4a-e*.

GICs were cultured on laminin (Sigma Cat. #L2020) coated tissue culture plates at 37°C, 5% $CO_2$. Cells were maintained in Neurocult media (Stem Cell Technologies Cat. #05751) supplemented with 1% penicillin/streptomycin solution (Sigma Cat. #P4458), Heparin (Stem Cell Technologies Cat. #07980), EGF (Peprotech Cat. #AF-315-09-1MG) and FGF (Peprotech Cat. #AF-100-18B-50UG) and passaged once they reached 80–90% confluence. GICs from the HGCC cohort were cultured on Poly-L-Ornithine (Sigma Cat. #P3655) and laminin-coated plates and maintained in media termed HGCC Media (*Supplementary file 4a*), supplemented with EGF and FGF, and passaged in the same manner as GICs from our own cohort. Separate passages of the same GIC line were considered to be biological replicates.

Dura-derived fibroblasts were reprogrammed into EPSCs as previously published (*Wilkinson et al., 2019*; *Yang et al., 2017*), EPSCs were then further differentiated into iNSCs with a Gibco commercially available kit (*Vinel et al., 2021*) (Gibco Cat. #A1647801). iNSCs were cultured on GelTrex (Gibco Cat. #A1413302) coated tissue culture plates and maintained in Neural Expansion media (*Supplementary file 4b*) at 37°C, 5% $CO_2$. iNSCs were passaged once they reached 80–90% confluence.

HEK293T cells were cultured in adherent conditions and maintained in IMDM media (Gibco Cat. #12440061) supplemented with 10% foetal bovine serum (FBS) (Gibco Cat. #10500064) and 1% penicillin/streptomycin solution (Sigma Cat. #P4458) at 37°C, 5% $CO_2$. They were detached using 1× Accutase (Millipore Cat. SCR005) for 5 min and passaged at ratios from 1:5 to 1:50.

## iAPC and iOPC generation

Differentiation of iNSCs into iAPCs was adapted from published protocols (*Tcw et al., 2017*). Differentiation was initiated on day –1, by seeding dissociated iNSCs at $1.5×10^4$ cells/cm$^2$ density on GelTrex-coated plates in Neural Expansion media. On day 0 Neural Expansion media was changed to Astrocyte media (ScienCell: Cat. #1801) supplemented with 2% FBS, astrocyte growth supplement (AGS), and penicillin/streptomycin solution, provided with the media. From day 2 onwards, media was changed every 48 hr for 20–30 days. At 80–90% confluence, the cells were passaged back to the starting seeding density ($1.5×10^4$ cells/cm$^2$), or at an approximate ratio of 1:6. Cells were detached using Accutase, and always cultured in the same Astrocyte media on GelTrex. Cell pellets were collected at three different timepoints throughout the differentiation process, when cells were confluent and passaged; approximately at days 10, 20, and 30. Differentiated astrocytes could be cryopreserved using Astrocyte media supplemented with 10% DMSO, or commercially available SynthFreeze (Gibco, Cat. #A1254201).

iNSCs were differentiated into iOPCs using a published protocol (*Douvaras and Fossati, 2015*), commencing from iPSCs and achieving fully mature oligodendrocytes at 95 days. Here, the protocol was started from iNSCs, equivalent to day 8 of the Douvaras-Fossati protocol, which were induced into OPCs up to day 75, when the original authors reported emergence of immature O4$^+$ oligodendrocytes. On day 0 of our protocol (day 8 of the Douvaras-Fossati protocol) Neural Expansion media was removed, and iNSCs cultured in N2 media (*Supplementary file 4c*) with 100 nM RA and 1 µM smoothened agonist (SAG) added freshly each day. Media was then changed daily with fresh RA and SAG until day 4, at which point cells became over-confluent and were detached and placed in N2B27 Media (*Supplementary file 4d*) with freshly added RA and SAG. A series of scratches were made with a cell scraper vertically, horizontally, and diagonally across each well, and the contents of a single well transferred into eight wells of an ultra-low attachment 24-well plate (Corning Cat. #3473), with extra N2B27 media. Aggregates were then cultured in suspension for a further 8 days, with media changed every 48 hr. On day 12 of the protocol N2B27 media was replaced with PDGF media (*Supplementary*

*file 4e*), and aggregates were cultured for a further 10 days, with media replaced every 48 hr. On day 22 of the protocol aggregates were plated onto Poly-L-Ornithine and laminin-coated plates and were cultured adherently until day 67 of the protocol, with PDGF media changes every 48 hr.

Two independent differentiations of iAPCs and iOPCs from iNSCs were considered to be biological replicates. For iNSCs two independent differentiations from iPSCs were considered to be biological replicates.

### Proliferation assay

For the comparison of iNSCs and iAPCs cells were seeded at $2 \times 10^4$ cells per well of a 24-well plate (CytoOne Cat. #CC7682-7524), whilst for the comparison of GIC lines cells were seeded at $1 \times 10^4$ cells per well of a 24-well plate. Then, at selected timepoints (every 24 hr) starting from day 1 or day 2, cells from each well were individually detached using Accutase, then centrifuged individually (1200 rpm) at 4°C for 5 min. At least three wells were detached and counted at each timepoint to generate technical triplicates. After detachment and centrifugation, cell pellets were resuspended in 100 µL DPBS and 10 µL was mixed with 10 µL of Trypan Blue (Sigma Cat. T8154-100ML) and live cells counted using a haemocytometer.

### Invasion assay

Transwell inserts with 8.0 µm pores (Sarstedt Cat. #89.3932.800) were placed into wells of a 24-well plate and coated with 100 µL of GelTrex. A total of 100,000 cells were then seeded into the transwell insert in 200 µL and 700 µL of normal growth media was added to the bottom of the well. Cells were then incubated in normal growth conditions for 24 hr, at which point cells on the inside of the transwell were removed using a cotton bud dampened with DPBS. Once cells inside the transwell were removed, cells on the bottom of the transwell were fixed using methanol, pre-chilled to –20°C, for 5 min at room temperature. After fixation, the bottom of the transwell was washed twice for 5 min using DPBS. The membrane of the transwell was then cut out and mounted onto a microscopy slide with mounting media including DAPI (Vectorlabs Cat. #H-2000). Transwell membranes were then analysed at the microscope and five representative images of nuclei on each membrane captured. For each biological group and replicate (different passages of cell lines), three technical replicate membranes were imaged. Finally, the number of whole nuclei in each image field were counted, using ImageJ software (Version 1.51m9), to ascertain how many cells migrated across the membrane.

### Neurosphere extreme limiting dilution assay

On day 0 of the assay, cells were seeded at a maximum cell density of 25 cells per well of a round bottom ultra-low attachment 96-well plate (Corning Cat. #7007). Cells at this density were then serially diluted 1:2 a total of four times to give five cell densities in total – 25, 12.5, 6.25, 3.125, and 1.5625 cells per well. For each biological replicate (different passages of cell lines), 12 technical replicates of each cell density were performed (12 separate wells). Cells were then incubated for 14 days with media changes every 48 hr, after which time the presence of neurospheres was assessed and counted for each well. To be considered a neurosphere, cells had to form 3D spherical clusters with smooth and defined edges and had to be greater than two cells in size. Results were analysed using the extreme limiting dilution analysis tool (*Hu and Smyth, 2009*), where the log proportion of negative cultures is plotted against the number of cells seeded, with a trend line indicating the estimated active stem cell frequency. The statistical significance of the differences between the estimated active stem cell frequencies of different cell lines was also tested using a chi-squared test as part of the analysis tool (*Hu and Smyth, 2009*).

### Animal procedures

All procedures were performed in accordance with licenses held under the UK Animals (Home Office Guidelines: animals Scientific Procedures Act 1986, PPL 70/6452 and P78B6C064 Scientific Procedures) Act 1986 and later modifications and conforming to all relevant guidelines and regulations. Orthotopic xenografts were performed on 8- to 12-week-old NOD SCID CB17-Prkdcscid/J mice (purchased from The Jackson laboratory) under anaesthesia with isoflurane gas and $5 \times 10^5$ primary human GIC in 10 µL PBS were slowly injected with a 26-gauge Hamilton syringe needle into the right cerebral hemisphere with the following coordinates from the bregma suture: 2 mm posterior, 2 mm

lateral, 5 mm deep, 10° angle. After the injection, scalps were cleaned with ethanol swab to remove any remaining cells and sutured with 4-0 Coated Vicryl Suture (Ethicon). After the surgery, mice recovered on a heatmap until they were fully awake. For the 5 days following surgery, mice were checked twice a day, then once a day and body weight was monitored once a week. Mice were kept on tumour watch until they developed brain tumour clinical symptoms and were then euthanised by neck dislocation and brains were harvested for histology analysis.

## DNA and RNA extraction

Cells used for qPCR analysis were pelleted and frozen at –80°C before RNA extraction. RNA was extracted by following the standard protocols of either Qiagen RNeasy Mini or Micro kits (Cat. #74104/74004). Some cell pellets were processed using Norgen Biotek RNA/DNA/Protein Purification Plus Kit (Cat. #47700), which allows genomic DNA, total RNA, and protein to be isolated from a single sample. RNA to be sent for RNAseq, and DNA to be sent for DNA methylation array, were prepared using the Norgen Biotek kit, according to the manufacturer's instructions.

## Reverse transcription and qPCR

Reverse transcription reactions were carried out by first mixing 1 µL random primers (Invitrogen Cat. #48190011), 500 ng RNA and ddH$_2$O up to 10 µL, primers were then annealed by heating to 65°C for 5 min, then 4°C for 5 min. Then 4 µL 5× FS Buffer, 1 µL 0.1 M DTT, 0.5 µL SuperScript III Reverse Transcriptase (Invitrogen Cat. #18080044), 1 µL 10 mM dNTP mix (Invitrogen Cat. #18427-013), and 3.5 µL ddH$_2$O were added to this reaction mixture.

qPCRs were carried out using Applied Biosystems Syber Green qPCR Master Mix (Cat. #4309155). Each reaction contained 2 µL of 2.5 ng/µL cDNA (5 ng total), 0.48 µL of 10 µM forward and reverse primer mix, 3.52 µL ddH$_2$O, and 6 µL of Syber Green Master Mix (12 µL total reaction volume). All qPCRs were run on an Applied Biosystems 7500 Real-Time PCR System or StepOnePlus Real-Time PCR System. A full list of primers and their sequences used throughout this project can be found in *Supplementary file 4f*.

## Flow cytometry and FACS

Primary and secondary antibodies used for flow cytometry and fluorescent activated cell sorting (FACS) staining are listed in *Supplementary file 4g*. Samples stained with unconjugated primary antibodies were incubated with species reactive secondary antibodies with various fluorophores conjugated. Samples for flow cytometry analysis were analysed using a BD LSRII Analyser, samples for FACS were processed using a BD FACS Aria Sorter.

For FACS of live iOPCs, cells were dissociated by removing culture media, adding FACS buffer (1:200 BSA Sigma Cat. #A3912, 1:250 EDTA 0.5 mM Ambion Cat. #AM9262, in DPBS) and homogenising the cells into a single-cell suspension. As the culture contained large aggregates, the cell suspension was passed through a 100 µm size filter. Cells were then counted and centrifuged at 1500 rpm for 5 min at 4°C followed by dispensing 100 µL of a 5×10$^6$ cells/mL suspension per tube for staining. Prior to staining cells were incubated in anti-CD16/32 FcR block (diluted 1:200 in FACS buffer) for 15 min at 4°C, washed and then incubated with conjugated or unconjugated primary antibodies for 30 min at 4°C. Staining was carried out with either single antibodies or combinations of antibodies. Finally, cells were pelleted and resuspended in DPBS before FACS.

For flow cytometry analysis of cell samples, cells were harvested, resuspended in FACS buffer, and blocked with FcR blocker in the same manner as for FACS analysis. Extracellular antibodies were first incubated with samples for 30 min at 4°C, followed by washing and resuspension in a fixable viability dye diluted 1:200 in DPBS and incubated for 20 min at 4°C. After further washing, cells were then fixed for 20 min at 4°C in 4% PFA diluted in a 1:1 ratio with FACS buffer. For the staining of intracellular targets, after fixation the cells were permeabilised using methanol for 5 min at room temperature and incubated with antibodies diluted in methanol for 20 min at room temperature. Finally, samples were washed in FACS buffer and resuspended for analysis.

## Immunocytochemistry

All cells analysed via immunocytochemistry were washed in DPBS and fixed by treatment with 4% paraformaldehyde for 15 min, then washed in DPBS for 5 min, three times. Fixed cells were stored

at 4°C in DPBS until staining. Cells were then permeabilised and blocked followed by staining with primary antibodies. Primary antibodies and the dilutions used in this study can be found in *Supplementary file 4g*. After primary antibody incubation, overnight at 4°C or 3 hr at room temperature, samples were washed in DPBS, and stained with species reactive secondary antibodies conjugated to various fluorophores for 1 hr at room temperature. After washing once again in DPBS, sample slides were then mounted using Fluoroshield mounting media with DAPI and sealed using nail varnish.

## Immunoblotting

Cell samples used for protein extraction were first pelleted by centrifugation, at full speed, for 5 min, then snap-frozen using dry ice and stored at –20°C until extraction. Protein was extracted from cell pellets by resuspending pellets in RIPA lysis buffer (25 mM Tris-HCl pH 7.6, 150 mM NaCl, 1% NP-40, 1% sodium deoxycholate, 0.1% SDS) supplemented with protease inhibitors (Santa Cruz Cat. SC-24948A), samples were then left to incubate on ice for 30 min. After incubation samples were centrifuged, at full speed, for 15 min, at 4°C, and the supernatant collected.

Protein concentration was determined using BCA assay (Thermo Cat. #23227) performed as per the manufacturer's instructions. Concentration of protein samples was then determined by interpolating the absorbance values of the unknown samples with a standard curve of known protein concentrations.

An equal amount of protein was then loaded into a 4–12% acrylamide gel (Invitrogen Cat. #NP0322BOX). Proteins were separated in SDS-PAGE (Thermo Fisher Cat. #NP-0001) and blotted onto a nitrocellulose membrane (GE Healthcare Cat. #10600002). Membranes were then blocked with 5% non-fat milk (Santa Cruz Cat. #SC-2325) in Phosphate Buffered Saline-Tween (PBS-T) (0.1% Tween 20 [Sigma Cat. #P9416-100ML] in PBS) for 1 hr at room temperature and then incubated with primary antibodies at 4°C overnight. The *Supplementary file 4g* summarises primary antibodies and their dilutions used in this study. After incubation of primary antibody membranes were washed three times for 5 min in PBS-T and then incubated, at room temperature, with the species appropriate peroxidase-conjugated secondary antibody at a dilution of 1:5000 for 1 hr. Membranes were further washed three times for 5 min in PBS-T before being visualised using ECL kit (GE Healthcare Cat. #RPN2232) and ChemiDoc Imaging System (Bio-Rad). Quantification of the protein expression was measured by densitometric analysis performed with ImageJ software (Version 1.51m9).

## Enzyme linked immunosorbent assay

Quantification of secreted proteins and chemokines such as interleukin-6 (IL-6) was performed using an enzyme linked immunosorbent assay (ELISA). Cells were cultured, prepared, and treated as already described, and after treatment growth media supernatant was collected for analysis by ELISA. Upon collection, supernatant was filtered using 0.22 µm syringe filter (Santa Cruz Cat. #SC-358812) and snap-frozen using dry ice, followed by storage at –80°C until analysis. The ELISA was then performed as per the manufacturer's instructions (BD Bioscience IL-6 ELISA Cat. #555220), and concentrations determined by interpolating absorbance values of samples using a standard curve.

## shRNA lentivirus production and transduction of cell lines

Lentivirus particles were produced using HEK293T cells. To produce shRNA lentivirus, a Lipofectamine 3000 (Invitrogen Cat. #L3000-015) transfection protocol was used. Transfection was carried out as per the manufacturer's standard protocol, by forming DNA-lipid complexes which were then incubated on cells for 6 hr followed by addition of packaging media for lentivirus harvesting. Packaging media consisted of Optimem reduced serum media supplemented with Glutamax (Gibco Cat. #51985-034), with 1 mM sodium pyruvate (Gibco Cat. #11360-039) and 5% FBS. Twenty-four hr post transfection, media was harvested from HEK293T cells and stored at 4°C, 10 mL of fresh packaging media was added. Fifty-two hr post transfection media was harvested once again and mixed with media previously harvested. Harvested supernatant was then centrifuged at 2000 rpm for 5 min to remove cell debris, then filtered through a 0.44 µm filter (VWR Cat. #514-0329). To precipitate and concentrate lentiviral particles, 5× polyethylene glycol (PEG) (Sigma #89510-1KG-F) was prepared by mixing 200 g PEG, 12 g NaCl (Fisher Cat. #S/3166/60) and 1 mL of 1 M Tris (pH 7.5) (PanReac AppliChem Cat #A4263,0500) then ddH$_2$O added to a total volume of 500 mL. pH of PEG was further adjusted to 7.2, and then autoclaved before use. 5× PEG was then added to harvested supernatant to give a final

concentration of 1× and then incubated at 4°C overnight. The following day harvested supernatant mixed with PEG was centrifuged at 1500 g for 30 min at 4°C, supernatant was removed and spun again to remove excess supernatant. The lentiviral pellet was then resuspended in an appropriate amount of DPBS, aliquoted and stored at –80°C until use.

Harvested virus was titrated to determine the transforming units per mL (TU/mL) for each volume of virus used during titration. Cell lines were infected overnight with the stated multiplicity of infection (MOI) of lentiviral particles. The TU/mL that achieved 5–30% of fluorescent tag positive cells during lentivirus titration was selected for the calculation of the MOI. The following day after infection with lentivirus, media was changed, and the cells were left to recover. Once confluent the cells expressing the desired construct were then purified by Puromycin selection or FACS, if transduced with constructs with a Puromycin resistance gene or fluorescent tag.

Glycerol stocks of competent bacteria containing shRNA plasmids, for lentivirus production, were purchased from Horizon Discovery Dharmacon. To obtain plasmid for lentiviral production, a small of amount of glycerol stock was extracted and added to LB broth (Sigma Cat. #L3522) supplemented with the relevant antibiotic, and grown up in large liquid culture, overnight at 37°C. Following this, plasmid DNA was isolated and purified using the Qiagen Maxi Prep kit (Qiagen Cat. #12963). Details of each shRNA plasmid constructs can be found in *Supplementary file 4h*.

## DNA methylation data processing

DNA used for methylation arrays was extracted and prepared as described above. Two biological replicates of each patient-matched GICs, iNSCs, iAPCs, and iOPCs were sent for DNA methylation array. DNA was assayed on the Illumina Infinium Methylation EPIC array (over 850,000 probes). Raw data was imported into an R workspace (R Version 3.5.0) and all analysis therein performed using the RStudio environment (Version 1.1.453). Raw data from the array was first processed using the ChAMP (*Tian et al., 2017*; *Morris et al., 2014*) (Version 2.12.4) R package to remove any failed detections and flawed probes. Along with data, metadata was also imported and used to help perform analysis. After this initial processing, data were then further processed and normalised using the Subset-quantile within array normalisation (SWAN) algorithm (*Maksimovic et al., 2012*). All methylation data used for further analysis such as differential methylation, PCA, or heatmap and dendrograms were SWAN normalised data.

After initial processing of DNA methylation array data, as described above, DMRs and genes were identified. To do so, output data from the SWAN normalisation algorithm in the form of beta values were imported into an R workspace and different datasets merged into a single dataset with the only probes that were common to all samples. After initial data processing, the R package DMRcate (*Peters et al., 2015*) (Version 1.18.0) was then used to first identify DMRs and then the corresponding genes. The minimum number of contiguous (or consecutive) differentially methylated probes per region to be called a DMR was set to 6, the beta cutoff was set to 0.3, and as the lambda parameter (bandwidth) was set to 1000, the scaling factor (C) was set to 2 as per the authors' recommendations. Unless otherwise stated here, all other parameters were set to their respective default settings.

## RNAseq data processing

RNA used for RNAseq was extracted and prepared as previously detailed. RNA samples from two biological replicates of patient-matched iAPCs, iNSCs, and GICs was of sufficient quality and quantity (minimum 1000 ng total mass at concentration of 50 ng/μL with 260:280 and 260:230 ratios equal to ~2) as to perform Poly-A library preparation followed by sequencing using the Illumina HiSeq 4000 platform at 75 PE. For sequencing of miRNAs, total RNA extracted as already outlined was prepared using the NEBNext smallRNA kit for Illumina (E7330L). In some cases where insufficient quantities of RNA were isolated from single passages of cells, RNA from two passages were combined. iOPC samples yielded very low quality and quantity of RNA after extraction and therefore samples were not suitable to be prepared using the Poly-A library preparation method. Instead, two biological replicate iOPC samples, for each patient, were prepared using the SmartSeq2 library preparation method and sequenced using the Illumina HiSeq 4000 platform at 75 PE.

The raw RNAseq data generated was processed in multiple ways depending on the output of the analysis. FastQC (Version 0.11.5) (https://www.bioinformatics.babraham.ac.uk/projects/fastqc/) was used to perform quality control of the raw data, to check the Phred score, the GC content distribution,

and the duplication levels, then the TrimGalore tool (Version 0.4.5–1) (DOI 10.5281/zenodo.5127898) was used to remove low-quality reads (Phred score <20) and residual adapters. For the use of differential expression analysis raw data was processed using the STAR gapped alignment software (*Dobin et al., 2013*) (Version 2.7.0), which generated gene counts. The reference genome used for alignment was Ensembl GRCh38 (release 90). Following alignment, prior to differential gene expression analysis, further processing of the STAR output data was performed. Genes with a counts per million (CPM) value <1, in a minimum of half of the samples per group (i.e., GIC or iNSC) plus one, were removed. DEGs were identified using the R package: EdgeR (*Robinson et al., 2010*; *McCarthy et al., 2012*) (Version 3.24.3) with the thresholds that the minimum absolute log fold change (logFC) in gene expression was 2 and the false discovery rate was less than 0.01. EdgeR analysis was performed using two statistical tests provided by the package – the likelihood ratio test (glmFit) and the quasi-likelihood ratio test (glmQLFit), however all gene lists used in further downstream analysis were generated from the more conservative glmQLFit test, unless otherwise stated.

For the differential gene expression analysis of GICs from the HGCC cohort, microarray data that had been pre-processed (normalised and Combat batch adjusted) was used. Differential gene expression analysis was then performed in an R workspace using the RStudio environment and limma package (*Law et al., 2014*) (Version 3.40.6). Linear modelling was implemented by the lmFit function and the empirical Bayes statistics implemented by the eBayes function and DEGs with a p-value < 0.05, and logFC >0.5, were selected by the topTable function. Volcano plots depicting the logFC and statistical significance of DEGs, from analysis of both microarray data and RNAseq data, were generated using the R package EnhancedVolcano (Version 1.5.10).

To generate heatmaps, dendrograms and perform PCA (all detailed below), a different approach was used to process raw RNAseq data. Alignment was instead performed using pseudoalignment package Salmon (*Patro et al., 2017*) (Version 0.13.1), with the output being transcript expression level, which were then pooled to give gene level expression estimates expressed as transcripts per million (TPM) (*Durinck et al., 2009*; *Durinck et al., 2005*). This unit is normalised for library size and transcript length. The reference genome used for Salmon alignment was Ensembl GRCh38 (release 90).

## PCA, heatmaps, and dendrograms

PCA was performed on both methylation array data and RNAseq data to validate the sequencing data. Output data from the pseudoalignment package Salmon (TPM counts) were used for PCA of RNAseq data, and beta values from SWAN normalised data we used for PCA of methylation array data. Prior to PCA, data were filtered to include only the most variable methylation probes or genes, the exact number used for each figure is stated in the relevant figure legend. The R package NOIseq (*Tarazona et al., 2015*; *Tarazona et al., 2011*) (Version 2.26.1) was used to perform PCA and then the results plotted, using R packages ggplot2 (*Wickham, 2009*) (Version 3.1.1) (for two-dimensional plotting of two PCs) or plotly (Version 4.9.0) (for three-dimensional plotting of three PCs).

Heatmaps and dendrograms were generated using both DNA methylation and RNAseq data. The RNAseq data used was the output from the pseudoalignment package Salmon (gene expression data in units of TPM). DNA methylation data used was beta values from SWAN normalisation. RNAseq data was filtered prior to analysis by removing genes with a TPM value <1 and the number of samples in each group (i.e., GIC or iNSC) that must express each gene to half the group size plus one. After this initial filtering, only the most variable genes were used in the analysis, with the exact number stated for each figure. In general, the Euclidean clustering method and complete distance methods were used as part of the standard heatmap.2 function in R.

## Gene signature analysis

Gene signature analysis was performed using ssGSEA method. In order to calculate ssGSEA enrichment scores for our samples, the R package GSVA (*Hänzelmann et al., 2013*) (Version 1.30.0) was used. The gene level expression estimates output, expressed as TPM, from alignment using the Salmon (detailed earlier) were used as the input for this analysis. As well as gene expression data, gene lists or signatures were also required, and all gene lists/signatures were formatted as Ensembl gene IDs.

The oligodendrocyte gene signatures used here were taken from a published study by *McKenzie et al., 2018*. The ACS was manually generated by compiling multiple astrocytic gene signatures into one coherent gene signature. Astrocytic gene signatures were first found in the xCell bioinformatics tool (*Aran et al., 2017*): xCell is a tool used to de-convolute a sample composed of a mixture of cell types into its respective cell types, based on gene signatures curated and validated by the authors. Multiple data sources were used to generate gene signatures for as many cell types as possible, and the authors generated a gene signature for a given cell type from each data source, meaning more than one gene signature was generated for each cell type. For example, three data sources contained astrocyte expression data, and thus two astrocyte signatures were generated from each data source, meaning there were six signatures for the tool to use to de-convolute mixed samples. In this present study, we have taken these six astrocyte signatures used by the xCell tool and merged them into one coherent signature. The ACS and the oligodendrocyte signatures used in this study can be found in *Supplementary file 4i*.

## Motif analysis using Homer

Identification of enriched binding motifs in genomic regions was performed using Homer Hyper-geometric Optimization of Motif EnRichment (Version 4.11, 10-24-2019) (*Heinz et al., 2010*). The tool findMotifsGenome.pl was used to perform de novo search as well as to check the enrichment of known motifs, in the context of the latest human genome annotation (hg38). Homer searched significantly enriched motifs (p-value <0.05) with a length spanning a wide range of standard values (6, 8, 10, 12, 15, 20, 25, 30, 35, 40, 45, 50 bp) in a region of default size (200 bp) at the centre of each sequence. Following the Homer guidelines (http://homer.ucsd.edu/homer/index.html), the option -mask was used, to minimise the bias towards long repeats in the genome, and the maximum number of mismatches allowed in the global optimization phase has been set to 3, to improve the sensitivity of the algorithm. Other settings have been left as default, such as the distribution used to score motifs (binomial). Finally, a scoring algorithm assigned a ranked list of best matches (known motifs or genes) to each de novo motif, to inform the biological interpretation of the results.

## Image analysis

Whole-slide images (WSI) of immunostained sections of xenografts derived from GICs lines were analysed using QuPath (*Bankhead et al., 2017*). A machine learning-based pixel classifier was manually trained on a subset of WSI to detect tissue sections and vimentin immunostaining and create corresponding annotations. The trained pixel classifier was then applied to the whole set of WSI. Prior to detecting vimentin immunostaining using the pixel classifier, tissue annotations were eroded/shrunk by 35 μm to exclude tumours at the edge of the tissue sections.

Tumour core annotations were created by eroding/shrinking vimentin annotations by 10 μm to exclude any small and isolated staining as well as thin processes projecting from the tumour core. The annotation was then dilated/expanded to return the tumour core annotation to its original border size. Any annotations that were smaller than 10,000 μm² were discarded to exclude small islands.

Gross tumour annotations were created by dilating/expanding vimentin annotations by 75 μm to fill in gaps between vimentin staining close together to be considered as part of the gross tumour. These were then eroded/shrunk by 100 μm and then dilated/expanded by 25 μm to smoothen and return to the original border size. Any gross tumour annotations that did not contain a tumour core annotation were discarded.

The tumour's invasiveness index (II) (*Amodeo et al., 2017*), independent of tumour size, was calculated as:

$$Invasiveness\ index = \frac{Gross\ tumour\ area}{Tumour\ core\ area}$$

## scRNAseq data analysis

Two public scRNAseq datasets have been analysed: tumour samples from seven newly diagnosed GBM patients (*Pombo Antunes et al., 2021*, patients ND1-ND7) and from eight GBM patients (*Neftel et al., 2019*, patients MGH102, MGH105, MGH114, MGH115, MGH118, MGH124, MGH125, MGH126, MGH143), both generated by the 10x Genomics platform. The gene expression matrices were downloaded from https://www.brainimmuneatlas.org/, and GEO (accession number

GSE131928), respectively. The gene expression matrices were merged, data was normalised, highly variable genes were detected, and their expression was scaled, followed by PCA, using the Seurat R package (Version 3.2.3). To account for the batch effect between samples, the cellular PCA embedding values were corrected with the harmony R package (Version 1.0), using a diversity clustering penalty parameter (theta) of one. Theta controls the level of alignment between batches, with higher values resulting in stronger correction. Next, the first 20 harmony corrected PCA embeddings were included in Louvain clustering (resolution = 0.25) and UMAP dimensionality reduction, using Seurat. The identified clusters were annotated as myeloid cells, lymphocytes, endothelial cells, and oligodendrocytes, based on expression of cell type markers. The remaining group of clusters were annotated as cancer cells.

The lymphocyte cluster was disaggregated and re-clustered, using 10 harmony corrected PCA embeddings and resolution = 1. By using specific gene markers, the clusters were classified into B cells, regulatory T cells, proliferating CD8 T cells, NKT cells, naive T cells, interferon-response T cells, two clusters of CD4 T cells, two clusters of CD8 T cells, and two clusters of NK cells.

The myeloid cells were also disaggregated and re-clustered based on 30 harmony corrected PCA embeddings, and resolution = 1. We identified monocyte cluster, DC cluster, three clusters of microglia-derived TAMs (mg-TAMs), SEPP1-hi monocyte-derived TAMs (moTAMs), hypoxic moTAMs, IFN-response moTAMs, proliferating moTAMs, TAMs up-regulating heat shock protein genes (HSP TAM: HSPA1A, HSPA1B, HSP90AA1, HSPH1, HSPB1), one cluster specific for patient MGH105, as well as myeloid-cancer cell doublets.

The cancer cells were disaggregated and re-clustered using 20 harmony corrected PCA embeddings, and resolution = 1. One of the clusters, which was expressing macrophage markers, was removed from further analysis as a cluster of macrophage-cancer cell doublets. Additive module scores for the astrocytic (ACS) and OPC gene signatures defined in this study were calculated for each cancer cell, using the AddModuleScore function of Seurat. This function yields the average expression level of each gene signature, subtracted by the average expression of a control gene set. Then, each cell has been assigned the signature with the highest score (ACS, OPC Enrich-300, or OPC Spec-300). Alternatively, additive module scores were calculated for the six gene signatures described by *Neftel et al., 2019* ('MES1', 'MES2', 'AC', 'OPC', 'NPC1', and 'NPC2'). The six gene signatures were defined using the genes in Supplementary Table S2 of *Neftel et al., 2019*. Each cell was also assigned the signature with the highest from the six scores. For the gene set enrichment pseudo-bulk analysis of the cancer cells, the raw UMI counts were summed for all cancer cells per gene for each patient using the aggregateAcrossCells function from the scuttle R package (Version 1.0.4). The pseudo-bulk counts were normalised using the CPM method and log2-transformation by edgeR (Version 3.32.1). Gene set ssGSEA enrichment scores for the ACS and OPC signatures, as well as for the six gene signatures of *Neftel et al., 2019*, were calculated using the GSVAR package (Version 1.38.2).

Spearman's correlation coefficients have been calculated between ACS pseudo-bulk enrichment score divided by the mean between the two OPC scores per patient, and the percentages of the different cell populations per patient. Furthermore, Spearman's correlation analysis was performed between the percentages of cells assigned to the distinct gene signatures per patient and the percentages of the different cell subsets present per patient. The data from patients ND7 and MGH143 was not used in these analyses, as they only contain CD45$^+$ and CD45$^-$ sorted cells, respectively.

## Statistical analysis and graphs

All statistical analysis and generation of graphs was performed using GraphPad Prism 9 or R with appropriate R packages already mentioned. Parametric data are presented as mean ± standard deviation. $p < 0.05$ was considered statistically significant, with p-values $< 0.0332$, $< 0.0021$, $< 0.0002$, $< 0.0001$ represented with *, **, ***, ****, respectively. Further information of the statistical analysis of specific datasets is indicated in the figure legends.

All scatter plots, time-series plots, bar graphs and survival curves, and accompanying statistical tests were generated with GraphPad Prism 9 or R. Venn diagrams were produced using the R package VennDiagram (*Chen and Boutros, 2011*), for the comparison and visualisation of gene lists. Fisher's exact tests, used to test the significance of the overlap between Venn diagram categories, were calculated using an online statistics tool (https://www.socscistatistics.com/tests/fisher/default2.aspx) (no reference available). Upset plots showing the number of DEGs found in various patient comparisons

and the overlaps between patient comparisons were generated using the online tool Intervene Shiny App (https://asntech.shinyapps.io/intervene/) (no reference available).

## Acknowledgements

This work is funded by grants from Brain Tumour Research (Centre of Excellence award to SM), Cancer Research UK (C23985/A29199 programme award to SM), Barts Charity (MGU0447 programme grant to SM). Part of the study was funded by the National Institute for Health Research to UCLH Biomedical research centre (BRC399/NS/RB/101410 to SB). SB is also supported by the Department of Health's NIHR Biomedical Research Centre's funding scheme. We acknowledge the use of data generated by the TCGA Research Network: https://www.cancer.gov/tcga.

## Additional information

### Funding

| Funder | Grant reference number | Author |
|---|---|---|
| Brain Tumour Research Centre of Excellence QMUL | CoE QMUL | James Boot<br>Gabriel Rosser<br>Claire Vinel<br>Nicola Pomella<br>Loredana Guglielmi<br>Silvia Marino |
| Cancer Research UK | C23985/A29199 | Xinyu Zhang<br>Silvia Marino |
| Barts Charity | MGU0447 | Silvia Marino |
| National Institute for Health Research UCLH BRC | (BRC399/NS/RB/101410 | Yau Mun Lim<br>Sebastian Brandner |

The funders had no role in study design, data collection and interpretation, or the decision to submit the work for publication.

### Author contributions

James Boot, Conceptualization, Data curation, Formal analysis, Validation, Investigation, Visualization, Methodology, Writing – original draft, Project administration, Writing – review and editing; Gabriel Rosser, Data curation, Formal analysis, Validation, Investigation, Visualization, Methodology; Dailya Kancheva, Data curation, Formal analysis, Validation, Investigation, Visualization, Methodology, Writing – review and editing; Claire Vinel, Resources, Formal analysis, Investigation; Yau Mun Lim, Formal analysis, Investigation, Methodology; Nicola Pomella, Formal analysis, Investigation, Visualization; Xinyu Zhang, Formal analysis, Investigation; Loredana Guglielmi, Resources, Supervision; Denise Sheer, Supervision, Writing – review and editing; Michael Barnes, Conceptualization, Supervision; Sebastian Brandner, Resources, Supervision, Visualization, Methodology; Sven Nelander, Kiavash Movahedi, Resources, Supervision, Methodology; Silvia Marino, Conceptualization, Supervision, Funding acquisition, Writing – original draft, Project administration, Writing – review and editing

### Author ORCIDs

James Boot ![ORCID] http://orcid.org/0000-0002-5629-5026
Yau Mun Lim ![ORCID] http://orcid.org/0000-0002-8774-9537
Sven Nelander ![ORCID] http://orcid.org/0000-0003-1758-1262
Kiavash Movahedi ![ORCID] http://orcid.org/0000-0002-0826-4399
Silvia Marino ![ORCID] http://orcid.org/0000-0002-9612-2883

### Ethics

All procedures were performed in accordance with licenses held under the UK Animals (Home Office Guidelines: animals Scientific Procedures Act 1986, PPL 70/6452 and P78B6C064Scientific Procedures) Act 1986 and later modifications and conforming to all relevant guidelines and regulations.

Informed consent was obtained and ethical approval was available for the study (National Health Service, Health Research Authority, National Research Ethics Service 08/H0716/16 Amendment 1 17/10/2014).

## Decision letter and Author response
Decision letter https://doi.org/10.7554/eLife.77335.sa1
Author response https://doi.org/10.7554/eLife.77335.sa2

---

# Additional files

## Supplementary files
• Supplementary file 1. Differentially expressed genes (DEGs) (log fold change and false discovery ratio [FDR]) from comparisons between patient-matched pairs of glioblastoma initiating cells (GICs) and induced neural stem cells (iNSCs) for 10 patient pairs from this study.

• Supplementary file 2. Differentially expressed genes in the myeloid cell populations of the integrated dataset from newly diagnosed human GBM tumours of *Pombo Antunes et al., 2021*, and the 10× GBM tumours of *Neftel et al., 2019*.

• Supplementary file 3. Differentially expressed genes in the lymphocyte cell populations of the integrated dataset from newly diagnosed human GBM tumours of *Pombo Antunes et al., 2021*, and the 10× GBM tumours of *Neftel et al., 2019*.

• Supplementary file 4. Media recipes, primer sequences, antibody details, shRNA plasmid details and gene signature lists used in this study.

• Transparent reporting form

## Data availability
All data generated during this study are included in the manuscript and supporting files, including source files. Sequencing data are available at public repositories.

The following datasets were generated:

| Author(s) | Year | Dataset title | Dataset URL | Database and Identifier |
|---|---|---|---|---|
| Boot J, Marino S | 2022 | RNASeq data progenitors | https://www.ncbi.nlm.nih.gov/geo/query/acc.cgi?acc=GSE196418 | NCBI Gene Expression Omnibus, GSE196418 |
| Boot J, Marino S | 2022 | DNA Methyl progenitors | https://www.ncbi.nlm.nih.gov/geo/query/acc.cgi?acc=GSE196339 | NCBI Gene Expression Omnibus, GSE196339 |

The following previously published datasets were used:

| Author(s) | Year | Dataset title | Dataset URL | Database and Identifier |
|---|---|---|---|---|
| Vinel C, Rosser G, Guglielmi L, Pomella N, Constantinou M, Zhang X, Boot JR, Jones TA, Millner TO, Dumas AA, Rakyan V, Rees J, Lim YM, Thompson JL, Vuononvirta J, Assan TE, Aley N, Lin Y, Liu P, Sheer D, Merry CL, Marelli-Berg F, Brandner S, Marino S | 2021 | Comparative analysis of glioblastoma initiating cells and patient-matched EPSC-derived neural stem cells as a discovery tool and drug matching strategy [RNA-Seq] | https://www.ncbi.nlm.nih.gov/geo/query/acc.cgi?acc=GSE154958 | NCBI Gene Expression Omnibus, GSE154958 |

*Continued on next page*

*Continued*

| Author(s) | Year | Dataset title | Dataset URL | Database and Identifier |
|---|---|---|---|---|
| Vinel C, Rosser G, Guglielmi L, Pomella N, Constantinou M, Zhang X, Boot JR, Jones TA, Millner TO, Dumas AA, Rakyan V, Rees J, Lim YM, Thompson JL, Vuononvirta J, Assan TE, Aley N, Lin Y, Liu P, Sheer D, Merry CL, Marelli-Berg F, Brandner S, Marino S | 2021 | Comparative analysis of glioblastoma initiating cells and patient-matched EPSC-derived neural stem cells as a discovery tool and drug matching strategy [array] | https://www.ncbi.nlm.nih.gov/geo/query/acc.cgi?acc=GSE155985 | NCBI Gene Expression Omnibus, GSE155985 |
| Antunes AR, Scheyltjens I, Lodi F, Messiaen J, Antoranz A, Duerinck J, Kancheva D, Martens L, De Vlaminck K, Van Hove H, Hansen SS, Bosisio FM, Van der Borght K, De Vleeschouwer S, Sciot R, Bouwens L, Verfaillie M, Vandamme N, Vandenbroucke RE, De Wever O, Saeys Y, Guilliams M, Gysemans C, Neyns B, De Smet F, Lambrecht D, Van Ginderachter JA, Movahedi K | 2020 | Single-cell profiling of myeloid cells in glioblastoma across species and disease stage reveals macrophage competition and specialization | https://www.ncbi.nlm.nih.gov/geo/query/acc.cgi?acc=GSE163120 | NCBI Gene Expression Omnibus, GSE163120 |
| De Vlaminck K, Romão E, Puttemans J, Pombo Antunes AR, Kancheva D, Scheyltjens I, Van Ginderachter JA, Muyldermans S, Devoogdt N, Movahedi K, Raes G | 2021 | Single-cell multi-omic profiling of glioblastoma-associated myeloid cells | https://ega-archive.org/studies/EGAS00001004871 | Ega-archive, EGAS00001004871 |
| Nelander S, Johansson P | 2020 | Expression data from "A patient-derived cell atlas of glioblastoma informs precision targeting of the proteasome" (Johansson et al, Cell Reports, 2020) | https://www.ncbi.nlm.nih.gov/geo/query/acc.cgi?acc=GSE152160 | NCBI Gene Expression Omnibus, GSE152160 |

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
