## [Editor Report]

Overall, this paper represents a large amount of highly detailed work that identifies a hypomethylated subtype of glioblastoma that provides exciting foundations for further translational investigations.

---

## [Decision Letter]

**Decision letter after peer review:**

Thank you for submitting your article "Global hypo-methylation in a subgroup of glioblastoma enriched for an astrocytic signature is associated with increased invasion and altered immune landscape" for consideration by *eLife*. Your article has been reviewed by 3 peer reviewers, one of whom is a member of our Board of Reviewing Editors, and the evaluation has been overseen by a Reviewing Editor and Wafik El-Deiry as the Senior Editor. The reviewers have opted to remain anonymous.

Essential revisions:

1) Clarity of the piece, as raised by reviewers 1 and 3. Particularly Reviewer 3 – overall the manuscript was challenging to read and should be greatly clarified in general and according to specific comments from reviewers.

2) Identification and characterization of SPRX2 seem rudimentary (Reviewer 2) and more details should be provided.

3) More context would be necessary – for example, In the context of what we know about glioblastoma classification (Verhaak, etc), where does this subgroup belong? How does it affect patient outcome?

*Reviewer #1 (Recommendations for the authors):*

Glioblastoma is a devastating tumor type and is known to be comprised of various transcriptionally defined subtypes, which are of biologic and therapeutic relevance. Here the authors claim to have identified a new molecular subgroup with specific hypomethylation and an astrocytic gene signature, which is associated with increased invasion and macrophage infiltration. In general, this would be an interesting conclusion of relevance to the field. However, the approach seems convoluted, and explained at great and complex length, to the point where it is very difficult, even for a specialized reader, to understand. Greater clarity and simplification/brevity of the language throughout would greatly benefit this work. Nonetheless, there appear to be significant methodological issues which challenge the conclusions reached by the authors. Overall, it is not clear that the data support the concept of a new subgroup of glioblastoma.

1. How does the process of generating iNSCs affect the methylation status of the cells.

2. Figure 1 C the group of GSCs of interest should be marked for the reader.

3. The lack of strong transcriptional association with the hypomethylated group is confusing and the link to glial biology seems like cherry-picking – is there an ontological-based approach that could show this? Are there other analytical approaches that could determine transcriptome differences associated with the hypomethylated bias group?

4. For the in vivo studies do the migration patterns correlate with the traditional transcriptional categories (PN, MES etc), or are they something different – this is not clear.

5. If this is a new subgroup of glioblastoma, how does it fit in with existing models, and what would be the impact on treatment response, and survival association with specific driver mutations?

*Reviewer #2 (Recommendations for the authors):*

1) How closely is the methylation pattern in GIC lines reflecting the parent tumour? It would be interesting to know whether parent tumours of bias/enriched GIC lines share DMRs and/or ACS enrichment. Also, please clarify which cutoff value will separate hypomethylation bias from non-bias GICs or tumours and how robust this value is.

2) Figure 2, panel F-H: the difference in enrichment scores may be significant, but the effect size seems marginal. To better put this into context, the scale of the y-axis should start at 0 for all panels. It is also unclear to me why enrichment for astrocyte signatures was tested against all cell lines, but enrichment for oligodendrocyte signatures was only tested in iOPC and iNSC. The oligodendrocyte signatures should also be tested against iAPC to demonstrate specificity.

3) The authors identify SRPX2 as the target gene associated with hypomethylation bias/ACS-enriched GBM cells. The functional validation of SRPX2 is very limited though and includes only 3 in vitro assays (proliferation, invasion, sphere formation). It would be good to see a more comprehensive validation of SRPX2 including in vivo transplantation of at least one of the cell lines. Furthermore, does knockdown of SRPX2 affect expression of ACS markers?

4) Macrophage infiltration data relies only on scRNA-Seq analysis. This data may be biased by sample area or input cell numbers. It would be good to validate changes in immune cell content e.g. by histological staining of parent tumour samples from bias/enriched and non-bias GIC lines.

5) Figure S2: The immunofluorescence staining for GFAP and CD44 (panel B) looks like background staining and not a real signal. The images should be replaced with better ones.

*Reviewer #3 (Recommendations for the authors):*

I am very sorry but I got too much stuck in this manuscript and not fully understanding which comparison served which purpose. If there are only ten or eleven samples to analyse, the difference between groups needs to be considerable as the high-dimensionality of the data makes the power limited. At least on the gene expression analysis, this was apparently not the case. The authors went on to do many different comparisons of GICs with iNSCs, iAPCs and iOPCs to find a rationale behind the reduced methylation signature, but why each comparison is done and why not always the same samples are compared is not always clear. It is quite possible that the comparisons are entirely valid, but it needs more clarification. I tried several attempts but had too many issues with fully understanding the flow of the experiments. That is not to say the manuscript is not of interest, methylation on gliomas is a very powerful tool, and finding new subtypes is of high relevance.

Consider the variables: GICs, iNSCs, iASC and iOPC and these are split into hypo and non-hypo samples either with or without enrichment for an ASC signature. For some analysis, subgroups are subsetted to hypo-bias/enriched. Likely because of this subsetting, the various comparisons use different samples for input. Any comparison can be done paired or unpaired (not always stated), and I could not find whether there was any correction for multiple testing/false discovery rate.

Just look at the following paragraphs on page 24/25 of the manuscript:

"To identify genes that may contribute to the phenotypic characteristics of the bias/enriched GICs, and the tumours derived thereof, the following differential expression analyses were performed on a patient-by-patient basis: iAPC versus GIC, iAPC versus iNSC and iNSC versus GIC, with the aim to identify genes that play a role in the differentiation of iAPCs and are deregulated in GICs. Lists of DEGs for each patient comparison were then overlaid to find DEGs that were either unique or present in more than one comparison. DEGs present in both the iNSC versus GIC, and iNSC versus iAPC comparisons, were selected for further analysis as these DEGs could play a role in the differentiation of NSCs to astrocytes and because of the consequent similar expression level between GICs and iAPCs could indicate that GICs are ontogenetically linked to these progenitors.

We aimed at identifying genes that were differentially expressed in the two comparisons of interest and were at the same time part of the ACS (Figure 4 D). We focussed on DEGs that were specific to GICs 19, 31 and 44 and shared between all three. …"

If I try to tease out what is stated here: there are three comparisons, for two phenotypes for a very limited number of patients. There is no data on individual comparisons, no data on the overlap between comparisons, they just end up with three genes. The comparison includes sample 44, likely because this sample had a minor AC phenotype (but was not hypomethylated). Sample 44 can therefore have been assigned to either label. Because the sample size is so small, it is important that labels are always similar. The comparisons also did not include samples 17 and 30, likely because iNSC were created from only five samples. If so, the comparisons are made on a very limited number of patients. There are just so many comparisons, filters used and sample selections made that probably make sense, but I had great difficulty in judging the validity thereof.

---

## [Author Response]

Essential revisions:1) Clarity of the piece, as raised by reviewers 1 and 3. Particularly Reviewer 3 – overall the manuscript was challenging to read and should be greatly clarified in general and according to specific comments from reviewers.

We have simplified and clarified the text.

We have streamlined the terminology and we use it consistently across the manuscript:

When referring to GICs with and without a hypomethylation bias we now define these B-GICs (Bias-GICs) and nB-GICs (non-Bias-GICs) respectively.When referring to GICs that have both a hypo-methylation bias and astrocytic signature (ACS) enrichment and those without, we now define these BE-GICs (Bias-Enriched-GICs) and nBE-GICs (non-Bias-Enriched-GICs) respectively.

We have re-written the result section, where we explain our rationale for identifying DEGs that play a role in the observed phenotype and have added a dedicated figure summarising the strategy schematically.

2) Identification and characterization of SPRX2 seem rudimentary (Reviewer 2) and more details should be provided.

We strongly feel it is beyond the scope of this revision to perform in vivo experiments in mice, whereby GIC with silenced SRPX2 are injected into the brain of recipient mice; these experiments would require at least one year from the injection of the GIC (shSRPX2 and ctr) to the functional assessment given the long incubation time of primary, patient-derived GIC-driven xenografts. Unfortunately, an attempt to optimise the available anti-SRPX2 antibody on FFPE material has failed, which prevented the assessment of SRPX2 expression at the invasion front in xenografts derived from B-GICs and nB-GICs. However, we have leveraged scRNASeq datasets from human glioblastoma to obtain additional validation of our hypothesis of a potential role of SRPX2 in invasion of tumours with an hypomethylation bias/ACS enrichment. We now show an enrichment for SRPX2 expression in one of two main invasive niche in glial tumours, namely the perivascular niche (page 25, line 22-24). We acknowledge that these remain correlative data, which will require in vivo functional validation and we mention this explicitly in the manuscript, page 25, line 26-27, however they provide additional support to our conclusion that SRPX2 may play a role in mediating at least in part the more invasive properties of tumours with an hypomethylation bias and astrocytic signature enrichment.

3) More context would be necessary – for example, In the context of what we know about glioblastoma classification (Verhaak, etc), where does this subgroup belong? How does it affect patient outcome?

On reflection we agree the term subgroup was confusing, we argue that the hypomethylation bias identified here is a feature of a proportion of GBMs. However, it is not associated with a particular subgroup among those already identified, rather it seems to span previous subgroups – 3/4 BE-GICs are RTK1 and 1 is RTK2, whilst of the 6 nBE-GICs, 1 is RTK1, 3 are RTK2 and 2 are Mesenchymal. We have not found any survival differences between B-GIC patients and nB-GICs, a finding which is perhaps not surprising given the short survival of all patients with glioblastoma and the lack of survival differences found also for both transcriptionally based and epigenetic-based glioblastoma subgroups (Verhaak et al. Cancer Cell, 2010). However, identification of subsets of glioblastoma with defined epigenetic and molecular features is an essential first step for future personalised therapy.

Reviewer #1 (Recommendations for the authors):Glioblastoma is a devastating tumor type and is known to be comprised of various transcriptionally defined subtypes, which are of biologic and therapeutic relevance. Here the authors claim to have identified a new molecular subgroup with specific hypomethylation and an astrocytic gene signature, which is associated with increased invasion and macrophage infiltration. In general, this would be an interesting conclusion of relevance to the field. However, the approach seems convoluted, and explained at great and complex length, to the point where it is very difficult, even for a specialized reader, to understand. Greater clarity and simplification/brevity of the language throughout would greatly benefit this work. Nonetheless, there appear to be significant methodological issues which challenge the conclusions reached by the authors. Overall, it is not clear that the data support the concept of a new subgroup of glioblastoma.1. How does the process of generating iNSCs affect the methylation status of the cells.

This is indeed a crucial point, and in fact we went to great lengths to validate our iNSC as compared to endogenous NSC or other iNSC induced with other methods, both in human and mice. This work is now published in Vinel et al. Nature Communications, 2021, this publication is referenced in the main text of the manuscript.

2. Figure 1 C the group of GSCs of interest should be marked for the reader.

This is a very good recommendation and has been implemented.

3. The lack of strong transcriptional association with the hypomethylated group is confusing and the link to glial biology seems like cherry-picking – is there an ontological-based approach that could show this? Are there other analytical approaches that could determine transcriptome differences associated with the hypomethylated bias group?

The lack of a direct transcriptional association with the B-GICs is not unsurprising given that DNA hypomethylation does not always lead to up-regulation of gene expression (as compared to hypermethylation, which is strongly correlated with gene repression). However, we have revisited this analysis as suggested to determine if there are global transcriptional differences between the B-GICs and nB-GICs – and PCA of GICs shows that this is indeed the case (Figure 1—figure supplement 1 B). Therefore, we have now explained the concept better as there are transcriptional differences between B-GICs and nB-GICs, including the enrichment for an astrocytic signature.

We do not agree with the statement that “the link to glial biology is cherry picking”. We have shown multiple lines of evidence emanating from our work pointing at this link. Firstly, hypo-methylated loci from B-GICs are enriched for transcription factor binding sites associated with astrocyte/glial differentiation (Figure 2). Secondly, miRNAs differentially expressed in B-GICs have been linked to astrocyte/glial differentiation (Figure 1—figure supplement 1), and thirdly, differential gene expression between B-GICs and nB-GICs identifies up-regulation of RUNX2 and down-regulation of OLIG2 in B-GICs, which are TFs associated with astrocyte and OPC differentiation respectively (Figure 3).

4. For the in vivo studies do the migration patterns correlate with the traditional transcriptional categories (PN, MES etc), or are they something different – this is not clear.

The reviewer is raising a very good point, 3/4 BE-GICs are RTK1 and 1 is RTK2, whilst of the 6 nBE-GICs, 1 is RTK1, 3 are RTK2 and 2 are Mesenchymal. Therefore, we conclude that the migration patterns do not correlate with the traditional transcriptional subgroups.

5. If this is a new subgroup of glioblastoma, how does it fit in with existing models, and what would be the impact on treatment response, and survival association with specific driver mutations?

On reflection we agree the term subgroup was confusing, we now argue that the hypomethylation bias identified here is a feature of a proportion of GBMs. It is not associated with a particular transcriptional or epigenetic subgroup among those already identified (Sturm et al. Cancer Cell, 2012) but seems to span previous subgroups. We have not found any survival differences between B-GIC patients and nB-GICs, a finding which is perhaps not surprising given the short survival of all patients with glioblastoma and the lack of survival differences found also for both transcriptionally based and epigenetic-based glioblastoma subgroups (Verhaak et al. Cancer Cell, 2010).

All GICs in the study were derived from glioblastoma samples, where the most frequent genetic alterations found in IDH-wildtype glioblastomas, including mutations in the TERT promoter, 10q loss, 7p gain, EGFR amplification, EGFR vIII mutation and p53 gene mutations, had been tested for diagnostic purposes. We do not find any genetic alteration which is found only in B-GICs, although we would like to emphasize that our cohort is small, and this hinders any assessment of enrichment for defined genetic alterations. It could be interesting to carry out whole genome sequencing analysis to assess the genetic profile of B-GIC more comprehensively, however this is beyond the scope of this manuscript, and it is to be noted that the caveat of the small cohort size would remain.

Reviewer #2 (Recommendations for the authors):1) How closely is the methylation pattern in GIC lines reflecting the parent tumour? It would be interesting to know whether parent tumours of bias/enriched GIC lines share DMRs and/or ACS enrichment.

This is an interesting point, which we have addressed, and the results are shown in Figure 1H. We show that the bias towards hypo-methylation is observed, albeit less pronounced, when the parental tumours, rather than the GIC derived thereof, are used for the comparison with the iNSC. This difference may be due to the cellular heterogeneity within the tumour tissue, including cells of the tumour microenvironment, which may obscure the methylation bias of the tumour cells. However, the enrichment for the ACS is not observed when the parental tumour tissue is used for the comparative analysis (data not previously shown but now added as Figure 3—figure supplement 1 C and described on page 22). It is possible this is due to the lower quality of the RNA, as compared to the excellent DNA quality, obtained from FFPE material, which limit the transcriptome characterisation. It is also possible that the transcriptome is more dynamic and prone to change, and so for example whilst GICs pass on their epigenetic signature to daughter cells when they proliferate, the regulation of their transcriptome may not be equally inherited by the daughter cells. Finally, the cellular heterogeneity, including the tumour microenvironment, could also be obscuring the astrocyte signature enrichment in the parental tumour. We are discussing these points in the MS, page 29, paragraph 2.

Also, please clarify which cutoff value will separate hypomethylation bias from non-bias GICs or tumours and how robust this value is.

We observed that GIC from the HGCC cohort sit on a wide spectrum with regards to the extent of their hypo-methylation bias. For example, we find 16 GICs with 50% – 60% hypo-methylated DMRs, 10 with 60% – 70% hypo-methylated DMRs, 15 with 70% – 80% hypo-methylated DMRs, 9 with 80% – 90% hypo-methylated DMRs and 3 with >90% hypo-methylated DMRs. In our smaller cohort though, the number of GIC was too low to see this spectrum. Therefore, it could be argued that setting a binary threshold with regards to a cut-off value for separating B-GICs and nB-GICs would not always be informative. One could argue that GICs with >50% hypo-methylated DMRs have a bias towards hypo-methylation, as do GICs with >90% hypo-methylated DMRs – however these two groups could be quite different. We believe therefore that in many cases, as we have done in Figure 1F, it may be more informative to stratify GICs, particularly those from large cohorts, into a spectrum rather than a binary classification. Future studies may wish to focus on the cause and effect of the extent of hypo-methylation bias of GICs across the spectrum.

Where a threshold was required to classify B-GICs / BE-GICs into two distinct groups for the purposes of comparative analyses such as differential gene expression or differential methylation analysis, we adopted lenient thresholds to include as many GICs that could be classified as having a bias. Thus, we used a threshold of >60% for our own cohort where syngeneic comparisons were possible. This threshold is now clearly stated in the text, page 18, line 9.

Meanwhile, for the HGCC cohort, where syngeneic comparisons were not possible, which introduces some noise into the percentage of hypo-methylated DMRs, we used a more lenient threshold of >50% – whereby GICs were classified as having a hypo-methylation bias if they had >50% hypo-methylated DMRs in all 5 iNSC comparisons. These thresholds are now clearly stated in the text, page 19, line 7.

2) Figure 2, panel F-H: the difference in enrichment scores may be significant, but the effect size seems marginal. To better put this into context, the scale of the y-axis should start at 0 for all panels.

Thank you for highlighting this, it is a good point, and it has been implemented, now all scale of the y-axis starts at 0. This applies also to Figure 3 A and Figure 3—figure supplement 1 A therefore we have changed these figures accordingly too. However, we would like to point out that even if the effect size is small the differences were shown to be statistically significant (Figure 2F: One-way ANOVA, Figures 2G & H: Mann-Whitney T-Test), therefore the conclusions to be drawn by these graphs are sound.

It is also unclear to me why enrichment for astrocyte signatures was tested against all cell lines, but enrichment for oligodendrocyte signatures was only tested in iOPC and iNSC. The oligodendrocyte signatures should also be tested against iAPC to demonstrate specificity.

We would like to highlight that the iOPC were analysed separately from the iAPC due to a difference in library prep methods leading to a strong batch effect. Due to the low yield and quality of RNA isolated from FACS sorted iOPCs we could not use PolyA library prep methods and instead SmartSeq2 library prep methods had to be used. We observed a large batch effect in the data when merging with samples prepped via PolyA methods. We have previously tried to correct for this batch effect using ComBatSeq however we found that this did not remedy the problem. As we could not correct for the batch effect, we felt it would be inaccurate and misleading to analyse the cell types together.

However, to address the concerns of the reviewer we have created Author response image 1 which shows the enrichment scores for the oligodendrocyte signatures in our Poly-A library prepped samples as well as reference CNS samples. Therefore, whilst we cannot analyse the iOPC and iAPC together we do show here that the OPC signatures are specific to the OPC cell type and iOPCs from Garcia-Leon et al. show significantly higher enrichment scores than other cell types such as iAPCs, pAstros, iNSCs and iNeurons.

**Author response image 1. sa2fig1:** 

3) The authors identify SRPX2 as the target gene associated with hypomethylation bias/ACS-enriched GBM cells. The functional validation of SRPX2 is very limited though and includes only 3 in vitro assays (proliferation, invasion, sphere formation). It would be good to see a more comprehensive validation of SRPX2 including in vivo transplantation of at least one of the cell lines.

We strongly feel it is beyond the scope of this revision to perform in vivo experiments in mice, whereby GIC with silenced SRPX2 are injected into the brain of recipient mice; these experiments would require at least one year from the injection of the GIC (shSRPX2 and ctr) to the functional assessment given the long incubation time of primary, patient-derived GIC-driven xenografts. Unfortunately, an attempt to optimise the available anti-SRPX2 antibody on FFPE material has failed, which prevented the assessment of SRPX2 expression at the invasion front in xenografts derived from B-GICs and nB-GICs.

To further validate a potential impact of SRPX2 on invasion, we have analysed its expression across tumour regions using the Ivy GAP resource. We found that the expression of SRPX2 is highest in regions of microvascular proliferation and hyperplastic blood vessels, whilst it is lowest in the leading edge and infiltrating tumour (page 25, line 22-24). This is potentially very interesting given that these two areas represent the two main invasive niches in glial tumours, namely the perivascular spaces (page 25, line 24-25) and the white matter tracts (Giese and Westphal, Neurosurgery, 1996). To further investigate this, we identified the top 200 genes up regulated in regions of microvascular proliferation and hyperplastic blood vessels, relative to other tumour regions. We then used these 200 genes as a signature and scored single tumour cells from Antunes et al. Finally, we correlated single tumour cells score for this signature against the ACS and OPC signatures used in this study and found that the ACS had a positive correlation with the genes up regulated in regions of microvascular proliferation and hyperplastic blood vessels. The OPC signatures had a negative correlation.

We acknowledge that these remain correlative data, which will require in vivo functional validation and we mention this explicitly in the manuscript, page 29, line 35 – 36, however they provide additional support to our conclusion that SRPX2 may play a role in mediating at least in part the more invasive properties of tumours with an hypomethylation bias and astrocytic signature enrichment.

We have added these findings to the manuscript on page 25 and in Figure 4 M & N as well as Figure 4—figure supplement 1 F & G.

Furthermore, does knockdown of SRPX2 affect expression of ACS markers?

To investigate the effect of silencing SRPX2 on ACS markers, we have selected four genes – LRCC17, CLU, FABP7, TIMP2 – as these genes were highly expressed in BE-GICs and had a positive correlation with SRPX2 expression according to TCGA data available on Gliovis. We assessed expression of these markers via qPCR in SRPX2 knockdown lines (N = 6) (both U3118 from the HGCC cohort and GIC19 from our cohort). We show in Author response image 2 that SRPX2 knockdown had no consistent impact on the expression of the selected genes. Overall, given that SRPX2 is a secreted protein that plays a role in the extracellular matrix, and not a transcription factor for example, it is perhaps not surprising that it does not impact widely on the expression of ACS markers.

4) Macrophage infiltration data relies only on scRNA-Seq analysis. This data may be biased by sample area or input cell numbers. It would be good to validate changes in immune cell content e.g. by histological staining of parent tumour samples from bias/enriched and non-bias GIC lines.

This is a very good suggestion, which we have followed. We have stained FFPE tumour tissue from parental tumours corresponding to BE-GICs and nBE-GICs for CD68 – a known marker of macrophages and found that there is a trend for a higher percentage of CD68+ cells in tumours from BE-GICs (p = 0.0528). Although not reaching statistical significance probably because of the small numbers of tumours stained and/or the more generic nature of CD68 as a macrophage marker as compared of the cell subtypes which can be identified in the scRNASeq datasets, we have added the data to the manuscript (Figure 5—figure supplement 1 C) as it does provide some additional validation even if only as a trend.

5) Figure S2: The immunofluorescence staining for GFAP and CD44 (panel B) looks like background staining and not a real signal. The images should be replaced with better ones.

Images replaced with better ones as suggested. Now named Figure 2—figure supplement 1.

Reviewer #3 (Recommendations for the authors):I am very sorry but I got too much stuck in this manuscript and not fully understanding which comparison served which purpose. If there are only ten or eleven samples to analyse, the difference between groups needs to be considerable as the high-dimensionality of the data makes the power limited. At least on the gene expression analysis, this was apparently not the case. The authors went on to do many different comparisons of GICs with iNSCs, iAPCs and iOPCs to find a rationale behind the reduced methylation signature, but why each comparison is done and why not always the same samples are compared is not always clear. It is quite possible that the comparisons are entirely valid, but it needs more clarification. I tried several attempts but had too many issues with fully understanding the flow of the experiments. That is not to say the manuscript is not of interest, methylation on gliomas is a very powerful tool, and finding new subtypes is of high relevance.

We have simplified and clarified the text.

We have streamlined the terminology and we use it consistently across the manuscript:

When referring to GICs with and without a hypomethylation bias we now define these B-GICs (Bias-GICs) and nB-GICs (non-Bias-GICs) respectively.When referring to GICs that have both a hypo-methylation bias and astrocytic signature enrichment and those without, we now define these BE-GICs (Bias-Enriched-GICs) and nBE-GICs (non-Bias-Enriched-GICs) respectively.

We have re-written the result section, where we explain our rationale for identifying DEGs that play a role in the observed phenotype and have added a dedicated figure summarising the strategy schematically.

Consider the variables: GICs, iNSCs, iASC and iOPC and these are split into hypo and non-hypo samples either with or without enrichment for an ASC signature. For some analysis, subgroups are subsetted to hypo-bias/enriched. Likely because of this subsetting, the various comparisons use different samples for input. Any comparison can be done paired or unpaired (not always stated), and I could not find whether there was any correction for multiple testing/false discovery rate.

Full details of the parameters i.e., statistical models, FDR thresholds and logFC thresholds used in the differential analyses carried out, are described in the Materials and methods of the manuscript (page 12).

Just look at the following paragraphs on page 24/25 of the manuscript:"To identify genes that may contribute to the phenotypic characteristics of the bias/enriched GICs, and the tumours derived thereof, the following differential expression analyses were performed on a patient-by-patient basis: iAPC versus GIC, iAPC versus iNSC and iNSC versus GIC, with the aim to identify genes that play a role in the differentiation of iAPCs and are deregulated in GICs. Lists of DEGs for each patient comparison were then overlaid to find DEGs that were either unique or present in more than one comparison. DEGs present in both the iNSC versus GIC, and iNSC versus iAPC comparisons, were selected for further analysis as these DEGs could play a role in the differentiation of NSCs to astrocytes and because of the consequent similar expression level between GICs and iAPCs could indicate that GICs are ontogenetically linked to these progenitors.We aimed at identifying genes that were differentially expressed in the two comparisons of interest and were at the same time part of the ACS (Figure 4 D). We focussed on DEGs that were specific to GICs 19, 31 and 44 and shared between all three. …"If I try to tease out what is stated here: there are three comparisons, for two phenotypes for a very limited number of patients. There is no data on individual comparisons, no data on the overlap between comparisons, they just end up with three genes. The comparison includes sample 44, likely because this sample had a minor AC phenotype (but was not hypomethylated). Sample 44 can therefore have been assigned to either label. Because the sample size is so small, it is important that labels are always similar. The comparisons also did not include samples 17 and 30, likely because iNSC were created from only five samples. If so, the comparisons are made on a very limited number of patients. There are just so many comparisons, filters used and sample selections made that probably make sense, but I had great difficulty in judging the validity thereof.

GIC44 was found to be enriched for the astrocytic signature, it did not have a hypo-methylation bias hence it was excluded from the two groups in our analysis because it did not meet the criteria of either. However, to improve the clarity of the manuscript we have removed GIC44 from the whole manuscript/analysis to avoid confusion. With regards to the confusion around the various differential analyses comparisons – we have included a schematic (Figure 4—figure supplement 1 A) dedicated to summarising the comparisons performed and the strategy we used to identify genes of interest as well as completely re-written the section of the manuscript describing the strategy and the reasoning behind the strategy as well as the results – we hope that this will be much clearer to follow.